# Inherent DNA-binding specificities of the HIF-1α and HIF-2α transcription factors in chromatin

James A Smythies[1,†], Min Sun[1,†], Norma Masson[1], Rafik Salama[1], Peter D Simpson[1], Elizabeth Murray[1], Viviana Neumann[1], Matthew E Cockman[2] [ID], Hani Choudhry[3], Peter J Ratcliffe[1,2,*] [ID] & David R Mole[1,**] [ID]

## Abstract

**Hypoxia-inducible factor (HIF) is the major transcriptional regulator of cellular responses to hypoxia. The two principal HIF-α isoforms, HIF-1α and HIF-2α, are progressively stabilized in response to hypoxia and form heterodimers with HIF-1β to activate a broad range of transcriptional responses. Here, we report on the pan-genomic distribution of isoform-specific HIF binding in response to hypoxia of varying severity and duration, and in response to genetic ablation of each HIF-α isoform. Our findings reveal that, despite an identical consensus recognition sequence in DNA, each HIF heterodimer loads progressively at a distinct repertoire of cell-type-specific sites across the genome, with little evidence of redistribution under any of the conditions examined. Marked biases towards promoter-proximal binding of HIF-1 and promoter-distant binding of HIF-2 were observed under all conditions and were consistent in multiple cell type. The findings imply that each HIF isoform has an inherent property that determines its binding distribution across the genome, which might be exploited to therapeutically target the specific transcriptional output of each isoform independently.**

**Keywords** DNA binding; HIF; hypoxia; transcription
**Subject Categories** Chromatin, Epigenetics, Genomics & Functional Genomics; Transcription

## Introduction

Transcriptional reprogramming of hypoxic cells enables a wide range of adaptive cellular responses that vary with the site, severity and duration of hypoxic stress [1–5]. A large body of work has implicated hypoxia-inducible factor (HIF) as the central transcriptional mediator of these responses [2]. Since tissue hypoxia is a complication of many human tissues, several strategies have been developed for the therapeutic modulation of different components of the HIF pathway [6–9]. However, the HIF transcriptional cascade has the potential to induce both adaptive and maladaptive responses [2], so that in many situations, the ideal approach would be one of the selective manipulations of specific components of the response. In that context, we have sought to better characterize factors governing the pan-genomic patterns of DNA binding of the two major isoforms, HIF-1 and HIF-2.

Hypoxia-inducible factor comprises an α/β heterodimer of basic helix–loop–helix PAS proteins [2,10]. HIF-α polypeptides are regulated by proteolysis, being targeted to the ubiquitin–proteasome pathway in oxygenated cells by the von Hippel–Lindau E3 ubiquitin ligase [2]. In contrast, HIF-β is constitutively expressed [11]. In hypoxia, HIF-α polypeptides escape destruction and are able to associate with HIF-β to drive transcriptional responses [4]. The two principal HIF-α isoforms, HIF-1α and HIF-2α (also known as EPAS-1), are widely expressed and form distinct heterodimers (HIF-1 and HIF-2) with the principal HIF-β isoform, HIF-1β [10,12]. HIF-1α and HIF-2α share a high degree of DNA sequence and structural homology, particularly within their DNA-binding and dimerization domains. However, an increasing body of data suggests that HIF-1 and HIF-2 heterodimers have distinct physiological functions and roles in disease [13,14]. For example, following the constitutive activation of both isoforms in VHL-defective kidney cancer, several pieces of evidence point to an oncogenic role for HIF-2α, whereas HIF-1α appears to manifest opposing tumour suppressor properties [15–17].

HIF-1 and HIF-2 manifest distinct patterns of tissue-specific expression [18,19] and different time courses [3,20] of induction by hypoxia. For instance, induction of HIF-1 peaks early following the onset of hypoxia, whereas activation of HIF-2 occurs more slowly and is more sustained [3,20]. HIF-1 and HIF-2 are also differentially responsive to non-hypoxic stimuli [21–23]. Despite a common consensus DNA-binding motif, HIF-1 and HIF-2 bind different but overlapping sets of sites in chromatin and transactivate only partially overlapping patterns of gene expression [24–26]. In some

1  NDM Research Building, University of Oxford, Oxford, UK
2  The Francis Crick Institute, London, UK
3  Department of Biochemistry, Faculty of Science, Center of Innovation in Personalized Medicine, King Fahd Center for Medical Research, King Abdulaziz University, Jeddah, Saudi Arabia
   *Corresponding author. Tel: +44 1865 612681; E-mail: peter.ratcliffe@ndm.ox.ac.uk
   **Corresponding author. Tel: +44 1865 613958; E-mail: david.mole@ndm.ox.ac.uk
   †These authors contributed equally to this work

settings, it has also been proposed that non-canonical association between specific HIF-α polypeptides and other transcriptional or signalling systems affects their output [27–29].

Although these findings have provided insights into the functional organization of the HIF system, a number of important questions remain unanswered. For instance, it is unclear how isoform-specific binding of HIF relates to the severity or duration of hypoxia, or how well HIF binding sites and their isoform specificities are conserved between cell types. In particular, the extent to which HIF isoforms compensate or compete at binding sites under conditions of differential expression is unclear. These questions are important not just in terms of understanding the physiology of responses to hypoxia, but also in assessing the potential of therapeutic approaches that target one or other HIF isoform specifically [30–32].

To address these points and related questions regarding factors constraining patterns of HIF binding to DNA, we used chromatin immunoprecipitation coupled to next-generation DNA sequencing (ChIP-seq) to examine pan-genomic patterns of HIF-1 and HIF-2 binding following varying degrees of severity and duration of hypoxic stimuli and in multiple cell types. The work reveals that both HIF-α isoforms bind chromatin in a stoichiometric ratio with HIF-1β, but with distinct binding distributions. These binding distributions appear to be inherent properties of each isoform and are largely unaffected by the degree or duration of hypoxia or by the presence or absence of the other HIF-α isoform. Isoform differences in patterns of HIF binding were conserved between cell types, despite different complements of binding sites in each cell type, suggesting that a conserved mechanism operates to distinguish HIF-1 from HIF-2 sites, which is independent of the mechanisms dictating cell-specific binding. Overall, our findings demonstrate that HIF-1 and HIF-2 bind to DNA in a largely distinct manner, irrespective of their relative level of activity, supporting the viability of HIF isoform-specific targeting as a therapeutic approach to selective modulation of the pathway.

# Results

## HIF-α binds chromatin in stoichiometric ratio with HIF-1β

Classically, HIF binds to chromatin as a canonical heterodimer of α- and β-subunits [24,33]. However, several reports have described non-canonical associations of HIF-α subunits with other transcription factors [27–29]. Moreover, under conditions of over-expression, patterns of HIF-α binding have been observed to deviate from those of HIF-1β, suggesting the potential for interactions with DNA other than through canonical α/β binding [16]. Thus, we first explored the concordance between endogenous HIF-α and HIF-1β binding in a range of settings. To obtain a robust dataset, ChIP-seq analyses of HIF-1α, HIF-2α and HIF-1β binding were performed in duplicate in HKC-8 renal tubule cells that had been cultured in 0.5% oxygen for 16 h. Sites that bound either HIF-1α or HIF-2α in both replicate samples were then identified using both the MACS and T-PIC peak callers [34,35]. To display the level of HIF-1β binding across HIF-α sites, the sites were then ranked according to HIF-1β signal intensity (counts per million, cpm), and heatmaps of HIF-α and HIF-1β signal intensity, centred on the summit of each peak ± 5 kb, were plotted (Fig 1A–D). These show clear HIF-1β signal above local background

intensity for both HIF-1α and HIF-2α sites, with no evidence of HIF-α binding at sites that do not also bind HIF-1β. As a separate check, HIF-1β signal intensity at all HIF-α binding sites was plotted against HIF-1β signal rank and compared to the mean HIF-1β signal intensity across non-HIF-α-bound enhancers, defined by DNA accessibility and the presence of histone H3K4me1 and H3K27ac marks [36] (Fig 1E and F). Virtually, all HIF-1α and HIF-2α sites had HIF-1β binding above levels seen at non-HIF-α-bound enhancers. The reciprocal analysis of HIF-1β sites also revealed HIF-α binding above background levels at all sites (Fig EV1A–C). However, it should be noted that these cells have not been stimulated to induce other binding partners of HIF-1β, such as the aryl hydrocarbon receptor, and under other conditions, we have seen evidence of HIF-1β binding without HIF-α. We then considered the possibility that the stoichiometric ratio of HIF-α binding to HIF-1β binding might vary quantitatively, rather than qualitatively, between sites. Total HIF-α signal (HIF-1α and HIF-2α combined) was first plotted against HIF-1β signal for all HIF-1α, HIF-2α or HIF-1β binding sites identified by the MACS peak caller (Fig 1G). Total HIF-α signal correlated strongly with HIF-1β signal. Overall, the ratio of total HIF-α to HIF-1β signal was tightly distributed, about a 1:1 ratio (Fig 1H), with differences in this ratio correlating very poorly with the same estimate in the other replicate (Fig 1I), as would be expected if apparent deviations from 1:1 stoichiometry were caused by noise in one or other dataset. Taken together, this suggests that the variation in the observed ratio of total HIF-α to HIF-1β signal between sites is a result of stochastic noise rather than systematic differences between sites.

These analyses, together with further similar analyses of endogenous HIF-α and HIF-1β binding in RCC4 and HepG2 cells (Figs EV2A–I and EV3A–I), demonstrated HIF-1β to be almost universally present at endogenous HIF-α binding sites with no clear evidence for variation in the stoichiometric ratio. Gene set enrichment analysis (GSEA), for all three cell lines, showed that, as for MCF-7 cells, the genes closest to each canonical HIF-α/β binding site were strongly enriched amongst upregulated but not downregulated genes (Fig 2A–C) indicating their functional significance. Thus for subsequent analysis, we focused on sites at which both HIF-α and HIF-1β signals were detected.

## HIF-1 and HIF-2 binding sites are independent of the duration or severity of hypoxia

We next examined whether the severity of the hypoxic stimulus could affect the binding distribution of either HIF-1 or HIF-2. HKC-8 cells were incubated in either 3 or 0.5% ambient oxygen for 6 h prior to harvest, and ChIP-seq analyses of HIF-1α, HIF-2α and HIF-1β binding were performed in duplicate. HIF-1α and HIF-2α protein levels both displayed graded induction in response to increasing severity of hypoxia, whilst HIF-1β expression was constitutive (Fig 3A). Canonical HIF-1 binding sites (identified by the MACS peak caller in both HIF-1α replicates and in both HIF-1β replicates) were identified at each oxygen concentration and combined to form a superset of sites that were identified at one or other or both oxygen concentrations. The HIF-1α signal intensity at 0.5% oxygen and 3% oxygen was then compared for each of these sites (Fig 3B). Overall, the HIF-1α signal intensity at 0.5% oxygen correlated strongly with that observed at 3% oxygen. The average signal intensity at 0.5% oxygen was greater than at 3% oxygen, consistent with

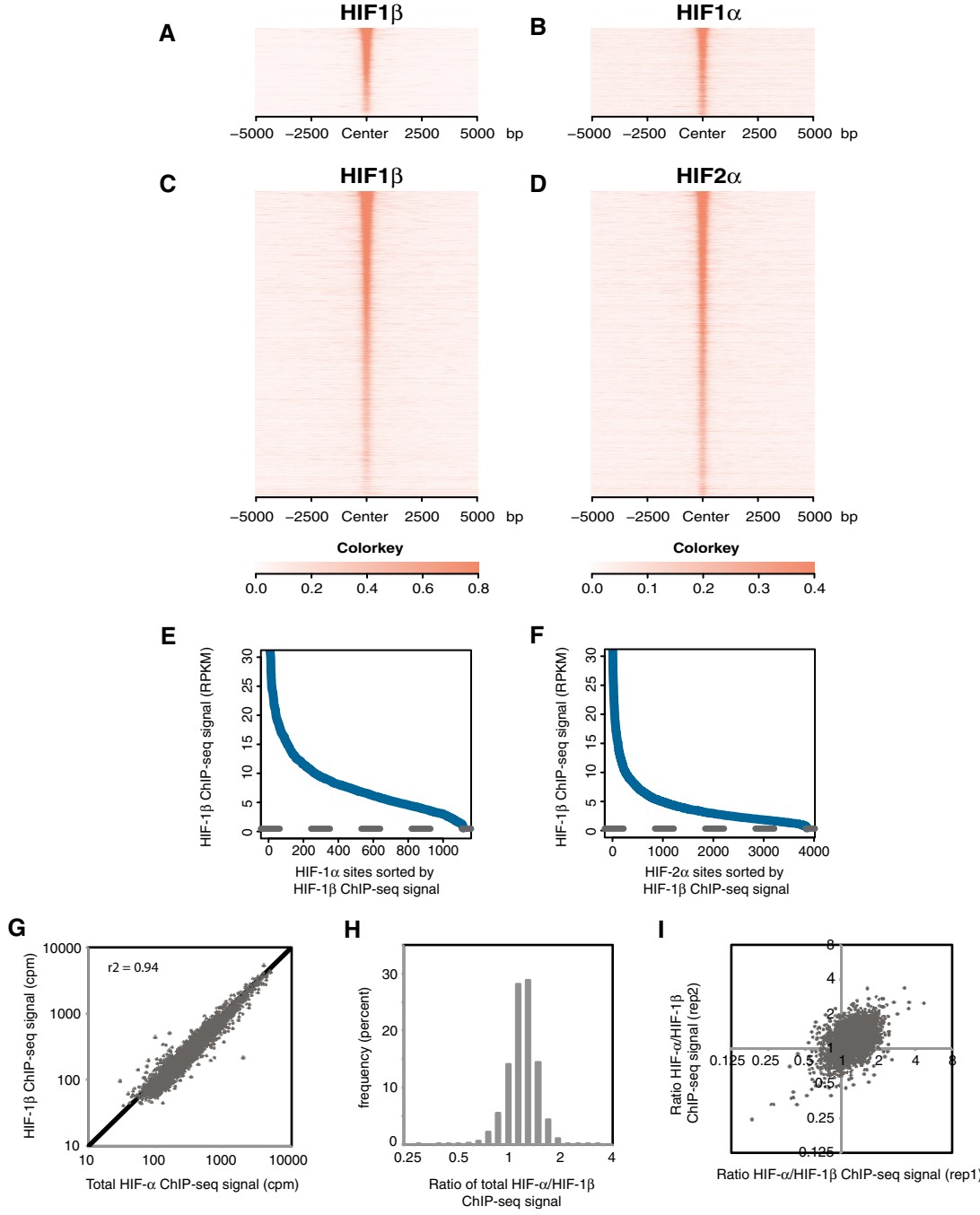

**Figure 1.  Stoichiometric binding of HIF-α and HIF-1β to chromatin in HKC-8 cells incubated in 0.5% atmospheric O₂ for 16 h.**

A–D    Sites that bound either HIF-1α (A and B) or HIF-2α (C and D) in both replicates were identified by the MACS peak caller and ordered on the *y*-axis according to HIF-1β signal intensity. Heatmaps show HIF ChIP-seq signal (read counts per million mapped reads, CPM; expressed as colour intensity, averaged across two independent ChIP-seq experiments) at HIF binding sites and across the flanking ±5kb regions (*x*-axis). (A) HIF-1β signal intensity and (B) HIF-1α signal intensity at HIF-1α binding sites and (C) HIF-1β signal intensity and (D) HIF-2α signal intensity at HIF-2α binding sites. HIF-1β signal intensity above local background levels is observed at all HIF-1α and HIF-2α binding sites.

E, F    Line plots showing average (*n* = 2, independent ChIP-seq experiments) HIF-1β signal intensity (solid blue lines) within the MACS defined (E) HIF-1α and (F) HIF-2α binding sites compared to the average background HIF-1β binding signal (dashed black line) at non-HIF-α binding accessible sites (defined by FAIRE-seq). Sites are ranked on the *x*-axis according to HIF-1β signal. HIF-1β signal intensity at both HIF-1α and HIF-2α binding sites was consistently above genome-wide background levels.

G    HIF-1β signal intensity was plotted against total HIF-α (HIF-1α + HIF-2α) signal intensity for all sites that bound one or more HIF subunits. A strong correlation was observed between HIF-α and HIF-1β signal intensities.

H    The ratio of total HIF-α signal to HIF-1β signal (*x*-axis) was determined for each site and plotted as a frequency distribution (*y*-axis) showing a tight unimodal distribution.

I    The non-reproducibility of the total HIF-α/HIF-1β ratio was assessed by plotting the ratio in replicate 1 versus that in replicate 2.

the higher levels of HIF-1α protein observed at the more severe level of hypoxia. In particular, there was no evidence that sites bound exclusively at one oxygen concentration, but not the other (i.e. sites were grouped close to the line of identity). Similar results were observed when the analysis was repeated for canonical HIF-2 binding sites (Fig 3C). HIF-1β signals at these sites mirrored those of HIF-α, consistent with the canonical binding observed above (Fig 3D). Thus, in HKC-8 cells, more severe hypoxia leads to increased HIF-1 and HIF-2 binding, commensurate with higher total protein binding, but does not qualitatively alter the distribution of this binding.

Nevertheless, it is possible that there are quantitative differences between sites with some loading progressively as the severity of hypoxia is increased, whilst others become saturated at milder degrees of hypoxia. We therefore examined the ratio of total HIF-α signal at 0.5% compared to 3% ambient oxygen at all sites that bound HIF at either oxygen concentration (Fig 3E, purple line). These values distributed around a single peak, but spanned a range of approximately eightfold. To determine whether the extremes of this range represented true biological differences or were the result of noise in one or the other ChIP-seq dataset, we compared the behaviour of HIF-α and HIF-1β signal, arguing that genuine biological differences in binding should be reflected in both datasets. Accordingly, we divided sites into an upper tertile, in which HIF-α signal loaded progressively (Fig 3E, solid red line) and a lower tertile, in which HIF-α signal was apparently saturated early (Fig 3E, solid blue line). We then examined the ratio of HIF-1β signal at 0.5 and 3% ambient oxygen in these two sets of sites (Fig 3E, dotted lines). The ratio of HIF-1β signals was significantly higher in the upper tertile group than in the lower tertile set. Thus, in HKC-8 cells, sites defined as either progressive or early loading manifest discernable overlap whether they were defined by the behaviour of HIF-α or HIF-β, indicating that at least some of differences were a biological reflection of heterodimeric binding rather than uncorrelated noise in the ChIP-seq assays. Examination of these groups of sites revealed no differences in total loading, in gene ontology assignments (ingenuity pathway analysis), in base composition within or immediately flanking the HRE motif or in the enrichment of non-HRE motifs (MEME-ChIP). However, a higher proportion of progressively loading sites had more than one HRE motif when compared to early loading sites (75% versus 44% for HIF-1 sites and 52% versus 44% for HIF-2 sites). Overall, these analyses suggest that the majority of HIF binding sites loaded similarly as the severity of hypoxia increased, broadly in accordance with the total binding of that isoform. Nevertheless, a limited tendency of some sites to load earlier or later as hypoxia severity increases was observed.

Finally, as HIF-1 and HIF-2 bind distinct, and only partially overlapping, sets of sites, we examined whether the degree of hypoxia

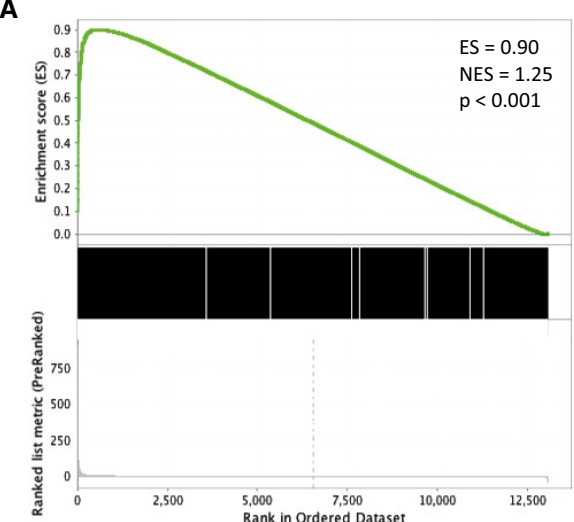

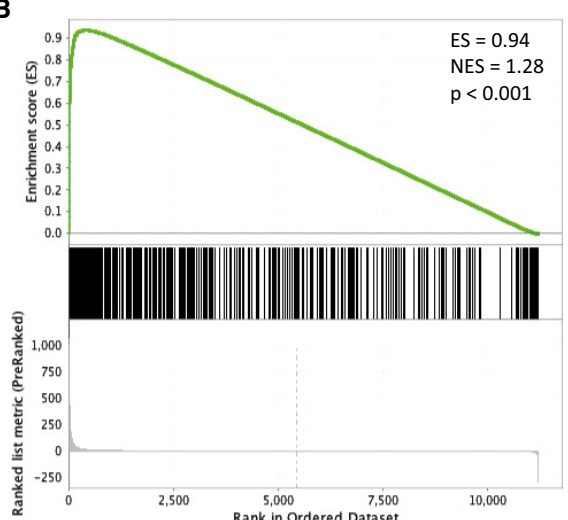

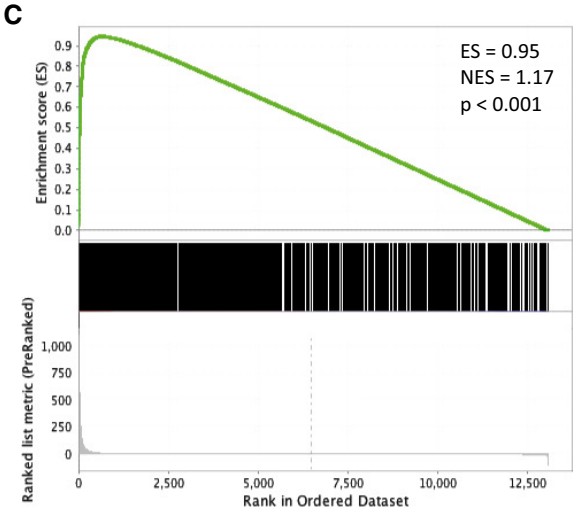

**Figure 2.  Gene set enrichment analysis (GSEA) showing hypoxic regulation of HIF-bound genes in (A) HKC-8, (B) RCC4 and (C) HepG2 cells.**

For each cell line, genes were ranked according to hypoxic induction (0.5% hypoxia for 16 h) in RNA-seq analyses (*n* = 3). Enrichment of genes (TSS) closest to each canonical HIF-α/β binding site [i.e. identified in both HIF-1α or both HIF-2α ChIP-seq experiments as well as both HIF-1β ChIP-seq experiments] was then examined using gene set enrichment analysis (ES, enrichment score; NES, normalized enrichment score).

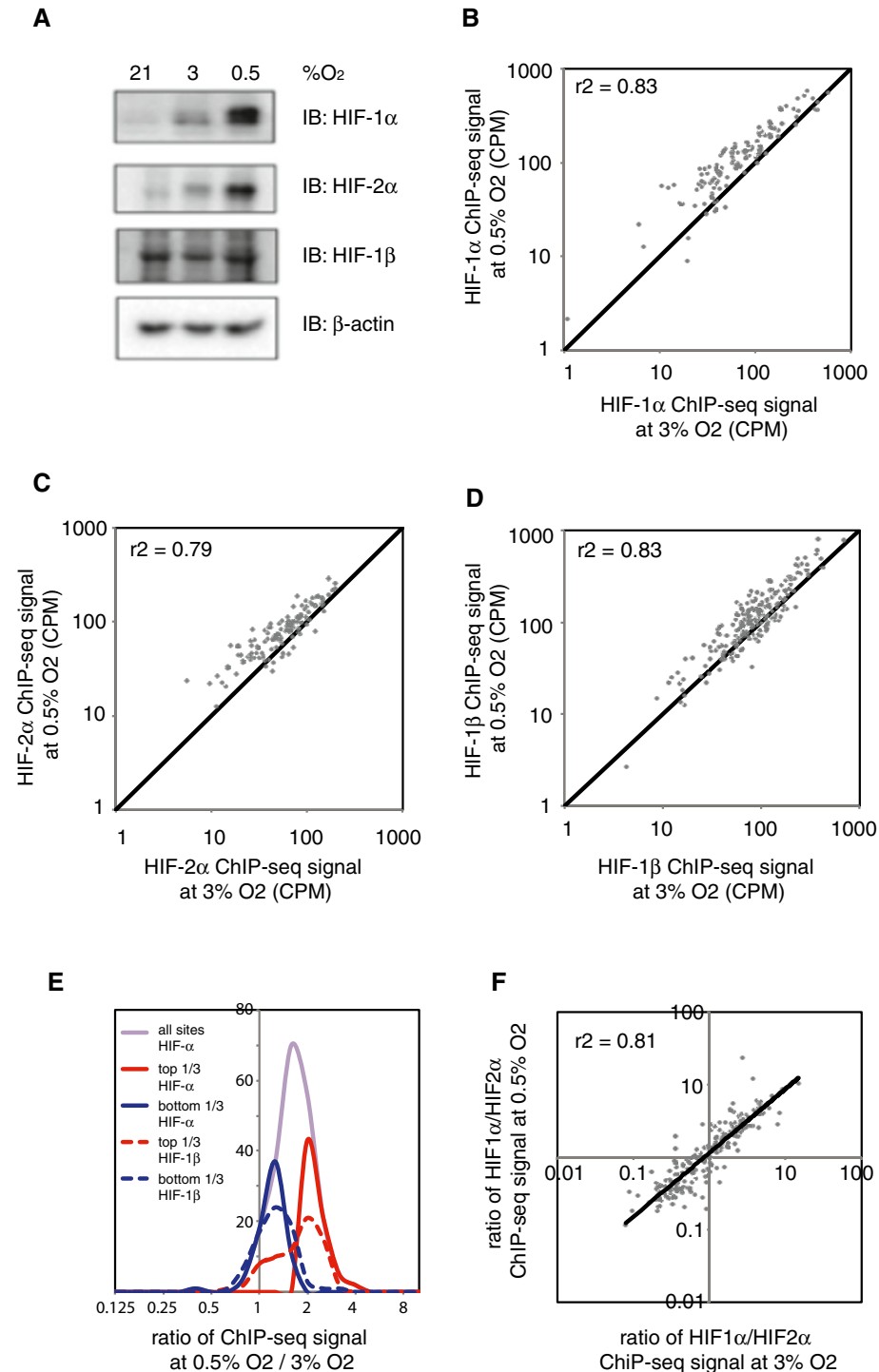

**Figure 3.  The effect of graded hypoxia on HIF binding.**
HKC-8 cells were incubated in 21, 3 or 0.5% ambient oxygen for 6 h.

A     Immunoblots show progressive induction of HIF-1α and HIF-2α protein levels with increasing severity of hypoxia.

B–D   (B) HIF-1α, (C) HIF-2α and (D) HIF-1β ChIP-seq signal intensities (averaged across two independent ChIP-seq experiments) at 0.5% hypoxia are plotted against those at 3% hypoxia for all canonical HIF binding sites that bound at one oxygen concentration or the other or both. ChIP-seq signal was increased at 0.5% oxygen, but correlated well with that at 3% oxygen and no new sites were generated.

E     Frequency distribution of ChIP-seq signals at 0.5% oxygen compared to 3% oxygen. The ratio of total HIF-α signal was unimodally distributed (purple line). Upper tertile HIF-α sites (solid red line) had a significantly ($P = 10^{-7}$, Wilcoxon rank sum test) higher ratio of HIF-1β signal (dotted red line) and vice versa (blue lines).

F     The ratio of HIF-1α to HIF-2α ChIP-seq signal for all canonical HIF binding sites at 0.5% oxygen was plotted against that at 3% oxygen. A strong correlation in the HIF-1α-to-HIF-2α ratio was observed between the two oxygen concentrations.

could alter the binding of each isoform to specific sites. The ratio of HIF-1α to HIF-2α was plotted for 0.5 and 3% ambient oxygen for all sites that bound either isoform at either oxygen concentration (Fig 3F). The HIF-1α-to-HIF-2α ratio at 0.5% oxygen correlated well with that at 3% oxygen, and no sites were observed to switch isoform specificity according to the degree of hypoxia.

We then determined the effect of duration of hypoxia on patterns of HIF binding. HKC-8 cells were incubated in 0.5% oxygen for 6, 16 or 48 h prior to harvest (Fig 4A). Again, strong correlations were observed between binding signal intensity across canonical HIF-1 and HIF-2 sites at 6 h and at 16 h (Fig 4B–D) and similarly between 6 and 48 h (Fig 4E–G). Average signal intensities were highest at 16 h, consistent with a transient rise and then fall in HIF-α protein levels of both isoforms by 48 h of hypoxia. There was no evidence that sites bound specifically at one time point, but not at another. Thus, as for oxygen concentration, varying the duration of hypoxia led to global quantitative differences in HIF binding, but did not qualitatively alter the distribution of HIF binding. Whilst some variation in the ratio of HIF-α signal at 48 h, compared to 6 h, was observed, in this case this was not mirrored by the ratio of HIF-1β signal (Fig 4H), suggesting that it resulted from random variation in the ChIP-seq signals, rather than a true biological difference.

As with the studies of graded hypoxia, the ratio of HIF-1α to HIF-2α at specific sites correlated well at different time points (Fig 4I and J), suggesting that in HKC-8 cells the ability of HIF-1 or HIF-2 to preferentially bind specific sites was independent of the duration as well as the severity of hypoxia, and a largely distinct property of each isoform.

## HIF-1α and HIF-2α do not compete for binding sites

To test the apparent independence of HIF-1 versus HIF-2 binding more directly, we next examined whether the ability of HIF-1 and HIF-2 to bind to specific sets of sites is affected by the presence of the other isoform. CRISPR-Cas9 was used to introduce frameshift mutations into the HIF-1α and HIF-2α genes that ablated production of each subunit in HKC-8 cells (Fig 5A). Cells were cultured in 0.5% ambient oxygen for 16 h. Ablation of HIF-2α was associated with a slight reduction in binding of HIF-1α, commensurate with slightly lower levels of HIF-1α protein (possibly a result of clonal variation) in these cells (Fig 5B). Ablation of HIF-1α had no effect on HIF-2α binding (Fig 5C). Importantly, in neither case did ablation of one HIF-α isoform increase binding of the other (i.e. there was no significant shift of signal at these sites above the line of equivalence).

Nevertheless, it is possible that ablation of one HIF-α isoform might increase binding of the other at a subset of sites; for instance, sites that specifically bound HIF-2α might do so because binding of HIF-2α excludes HIF-1α from binding or vice versa. Sites were therefore classified as HIF-1α specific or HIF-2α specific according to the ratio of HIF-1α to HIF-2α signal in wild-type cells. We then examined the effect of HIF-2α ablation on HIF-1α binding at HIF-2α-specific binding sites (Fig 5D). Comparison of HIF-1 binding in wild-type and HIF-2 defective cells provided no evidence for an increase in binding at these sites (compare red boxes, Fig 5D). Similarly, ablation of HIF-1α was not associated with any shift to HIF-2α binding at sites that bound HIF-1α in the wild-type cells (compare blue bars, Fig 5E). Remarkably in the HIF-α mutant cells, no significant increase in binding was observed for the remaining HIF-α isoform even at sites that bound to both isoforms in wild-type cells (Fig 5F and G). Furthermore, we did not observe any significant difference in the number of HRE motifs at shared sites compared to sites that bound only one isoform (Wilcoxon signed-rank test), any spatial separation of the HIF-1α and HIF-2α signals or any broadening of the HIF-1β peak that might have suggested that both isoforms bound concomitantly, but at different sites.

These findings suggested that HIF-1 and HIF-2 behave largely independently of each other in respect of the distribution of DNA binding across the genome. To analyse this further, we examined the effects of ablating either isoform on the distribution of binding sites for HIF-1 versus HIF-2 with respect to the transcriptional start sites at promoters. As observed previously, HIF-1 and HIF-2 differed markedly in this distribution, with HIF-1 binding more frequently close to and HIF-2 binding more frequently distant from promoters (Fig 6A–G). For both HIF-1 and HIF-2, this pattern remained unaltered when the other isoform was deleted, and was corroborated by HIF-1β, which distributed in a promoter-proximal or promoter-distal manner dependent on whether HIF-1α or HIF-2α remained intact (Fig EV4A–D).

Thus, the ability of HIF-1α or HIF-2α to bind specific sites is independent not only of the degree and duration of hypoxia, but also of the presence or absence of the other isoform, and appears to represent an inherent property of each isoform.

## Conservation of HIF-1 and HIF-2 binding sites between cell types

Given the evidence for the binding distributions of HIF-1 and HIF-2 being largely independent of each other, we next sought to compare the cell-type specificity of their binding patterns across

---

**Figure 4. The effect of duration of hypoxia on HIF binding.**
HKC-8 cells were incubated in 0.5% ambient oxygen for 6, 16 or 48 h.

A    Immunoblots show induction of HIF-1α and HIF-2α protein level, which is highest at 6 h and falls by 48 h.

B–D   (B) HIF-1α, (C) HIF-2α and (D) HIF-1β ChIP-seq signal intensities (averaged across two independent ChIP-seq experiments) at 16 h are plotted against those at 6 h for all canonical HIF binding sites that bound at any of the three time points. ChIP-seq signal was increased at 16 h of hypoxia compared to 6 h of hypoxia, but correlated well between the two time points, and no novel sites were observed at either time point.

E–G   The same plots comparing (E) HIF-1α, (F) HIF-2α and (G) HIF-1β ChIP-seq signals at 48 and 6 h of hypoxia. ChIP-seq signal at 48 h correlated well with that at 6 h, and again, no novel sites were observed at either time point.

H    Frequency distribution of ChIP-seq signals at 48 h compared to 6 h of hypoxia. The ratio of total HIF-α signal had a unimodal distribution (purple line). However, upper tertile HIF-α sites (solid red line) had a comparable ratio of HIF-1β signal (dotted red line) to the ratio of HIF-1β signal in the lower tertile HIF-α sites (blue lines), suggesting that differences in HIF-α signal resulted from random variation rather than a true biological difference.

I, J   The ratio of HIF-1α to HIF-2α ChIP-seq signal for all canonical HIF binding sites at (I) 16 h of hypoxia and at (J) 48 h of hypoxia was plotted against that at 6 h of hypoxia. A strong correlation in the HIF-1α-to-HIF-2α ratio was observed between the various time points.

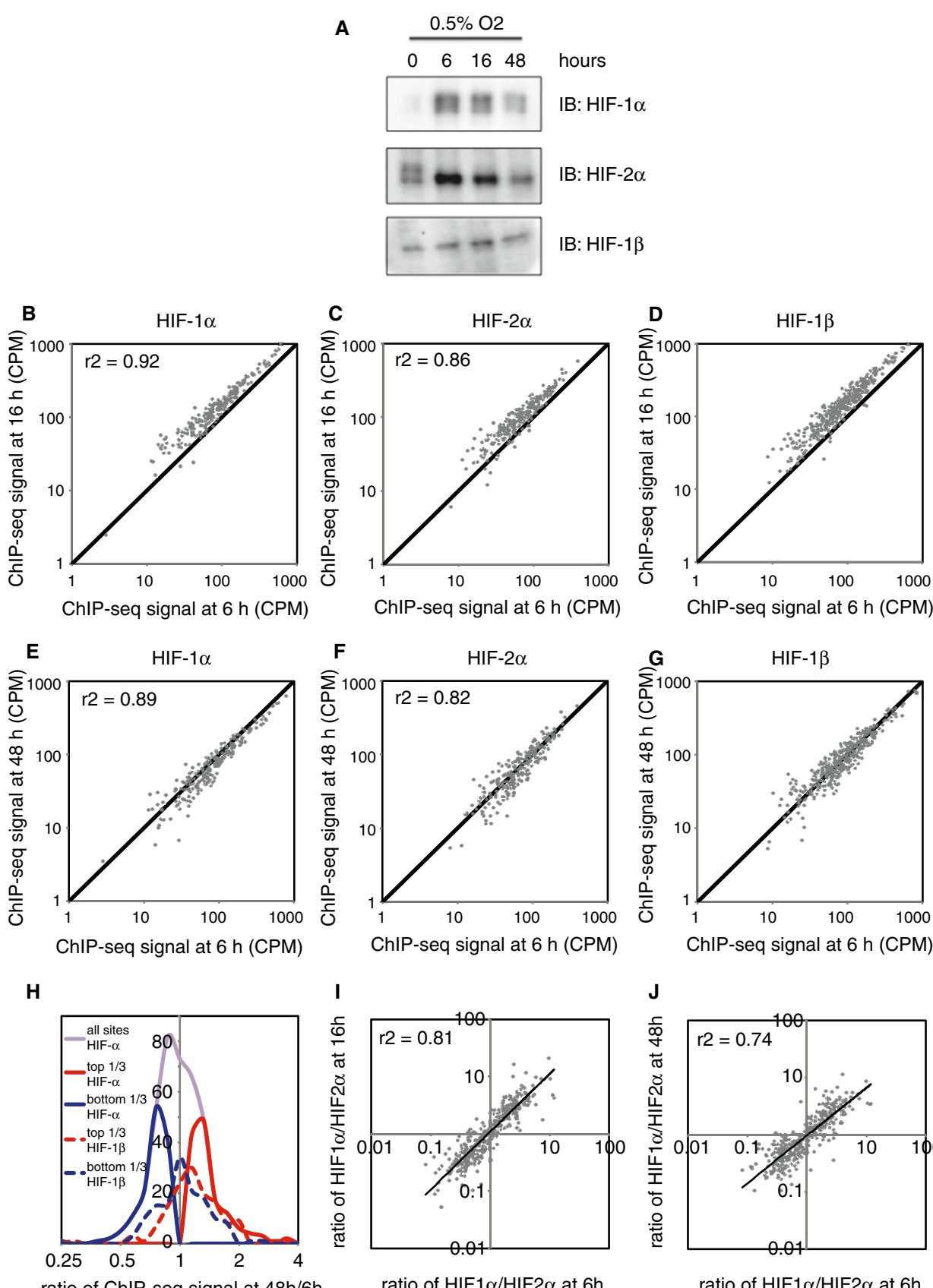

**Figure 4.**

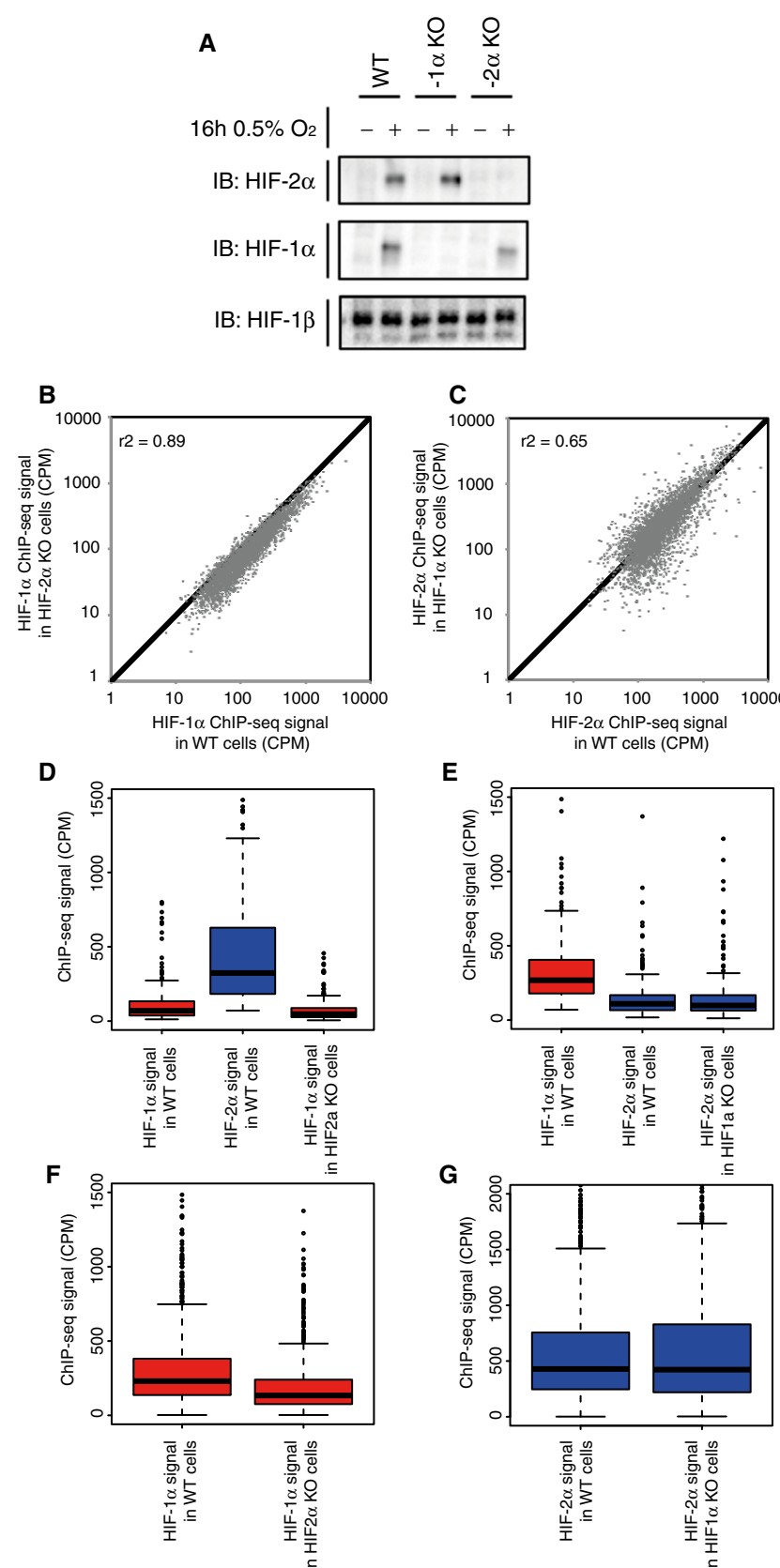

**Figure 5.**

**Figure 5. HIF-1α and HIF-2α do not compete for binding sites.**

The expression of either HIF-1α or HIF-2α was ablated in HKC-8 cells using CRISPR-Cas9.

A   Cells were incubated in 0.5% ambient oxygen for 16 h. Immunoblots show HIF-1α, HIF-2α and HIF-1β protein levels in the wild-type and single clones of CRISPR-Cas9 engineered cell lines.

B   HIF-1α ChIP-seq signal in the HIF-2α KO cells was plotted against that in the wild-type cells for all canonical HIF-1 binding sites in either cell type.

C   HIF-2α ChIP-seq signal in the HIF-1α KO cells was plotted against that in the wild-type cells for all canonical HIF-2 binding sites in either cell type. No significant increase (Wilcoxon signed-rank test) in binding of either isoform was observed following deletion of the other.

D   Box-and-whisker plots showing the effect of HIF-2α inactivation on HIF-1α ChIP-seq signal at sites that specifically bound HIF-2α, but not HIF-1α in wild-type cells showing a slight, but significant ($P = 4 \times 10^{-5}$, Wilcoxon signed-rank test) decrease rather than increase (compare red boxes) consistent with the reduction in HIF-1α protein observed in panel (A).

E   The effect of HIF-1α inactivation on HIF-2α ChIP-seq signal (compare blue boxes) at sites that specifically bound HIF-1α, but not HIF-2α in wild-type cells showing no significant effect ($P = 0.5$, Wilcoxon signed-rank test).

F, G   The effect, at sites that bind both isoforms in wild-type cells, of (F) HIF-2α inactivation on HIF-1α binding intensity showing a similar small but significant reduction rather than increase ($P = 2 \times 10^{-16}$, Wilcoxon signed-rank test) as above and of (G) HIF-1α inactivation on HIF-2α binding intensity showing no significant effect ($P = 0.4$, Wilcoxon signed-rank test).

Data information: In the box-and-whisker plots, the thick horizontal bars show the median, the boxes depict the interquartile range, and the whiskers represent the full range unless this is more than 2.5 times the interquartile range, in which case more extreme values are shown as individual points. ChIP-seq signal is averaged across two independent experiments.

cells with different levels of HIF-1α and HIF-2α protein (Fig 7A). ChIP-seq analyses of HIF-1α, HIF-2α and HIF-1β binding were performed in duplicate in HepG2 cells (incubated in 0.5% oxygen for 16 h) and normoxic RCC4 cells, in addition to the HKC-8 datasets above. In total, 1,807 canonical HIF-1 sites and 1,000 HIF-2 sites in HepG2 cells, 519 HIF-1 and 608 HIF-2 sites in RCC4 cells, and 1,080 HIF-1 and 3,240 HIF-2 sites in HKC-8 cells were identified by the presence of both HIF-α and HIF-1β isoforms in each of the two replicates. This, together with 356 canonical HIF-1 and 301 HIF-2 sites previously described for MCF-7 cells [24], provides a comprehensive analysis of HIF binding in four separate cell lines (Fig 7B and C). Overall, HIF-1 sites showed a higher level of conservation between different cell lines than HIF-2 sites, with approximately 25% of HIF-1 sites and 15% of HIF-2 sites being shared between two and more cell lines (Fig 7D). Conserved sites were generally more promoter-proximal than cell-type-specific binding sites (Fig EV5A–H).

Although this categorical definition of HIF binding allows comparisons between the behaviour of HIF-1 and HIF-2, it does not allow assessment of quantitative aspects of cell-type-specific binding and may exaggerate differences between sites which are just above or below "peak calling" thresholds. To address this, we performed quantitative analyses of pair-wise comparisons with the index (HKC-8) cell (Fig EV6A–H). As with analysis of heterodimeric binding for all sites ascertained in HKC-8 cells, we plotted HIF binding intensity against rank in the comparator cell (Fig EV6G and H). Sites that were identified by the peak caller as binding only in HKC-8 cells had weak ChIP-seq signal in the "non-binding" cell lines, suggesting that categorization by peak caller tends to over-estimate cell-type specificity. However, signals were substantially lower in the "non-binding" cell lines, indicating a high level of quantitative specificity.

Finally, we wished to determine whether, despite cell-type differences in HIF binding sites, isoform-specific patterns of binding were preserved. First, we tested whether the different binding distributions of HIF-1 and HIF-2 with respect to transcriptional start sites are conserved between different cell types. These distributions are depicted in Fig 4 and reveal distinct profiles for HIF-1 (Fig 6A, C and E) versus HIF-2 (Fig 6B, D and F). Remarkably, despite individual binding sites differing greatly, the respective binding distribution

profiles for HIF-1 and HIF-2 remain similar across cell types. Interestingly, the distribution of accessible HRE motifs (as defined by the presence of a core HRE in open chromatin—Fig 6G) differs significantly from that of either HIF-1 or HIF-2 binding sites, suggesting that the distribution of both HIF-1 and HIF-2 binding sites is biased both towards and away from promoters by factors that appear to be common across cell types. Second, we sought to examine whether the ratio of HIF-1 to HIF-2 binding at specific sites was conserved between cell types. For these analyses, we defined sites that were shared between two cell types, measured the ratio of HIF-1α to HIF-2α binding and compared this between the two cells. This revealed a remarkably strong correlation between the HIF-1α:HIF-2α ratio in one cell type and that in the other cell type (Fig 7E–G). Thus although the sites vary between cell types, the isoform specificity of individual sites that are shared is conserved between different cell types.

### Epigenetic landscape at HIF-1 and HIF-2 binding sites

Given the distinct distributions of HIF-1 and HIF-2 across the genome, we also examined the association between canonical HIF-1 and HIF-2 binding sites defined in HepG2 cells (incubated in 0.5% oxygen for 16 h) with specific histone modifications, using publically available ENCODE [37] ChIP-seq data for eight histone modifications in the same cell line (Fig 8A). Consistent with their distinct binding distributions, HIF-1 associated more strongly with histone H3K4me3 modifications (a mark of regulatory elements primarily associated with promoters and transcriptional start sites [36]), whilst HIF-2 associated more strongly with H3K4me1 (a mark known to be associated with enhancers and other distal regulatory elements [36]). H3K4me2, which marks regulatory elements associated with both promoters and enhancers, was enriched at both HIF-1 and HIF-2 binding sites. H3K9ac (a mark of active regulatory elements with a preference for promoters [36]) was more strongly associated with HIF-1 binding sites than with HIF-2 binding sites, whilst H3K27ac (a mark of both active enhancers and promoters [36]) was more strongly enriched at HIF-2 binding sites. Interestingly, H3K27me3, which affects polycomb repression of regulatory domains [38], also showed weak enrichment, particularly at HIF-1 binding sites. Overall, the pattern of histone modifications at HIF-1

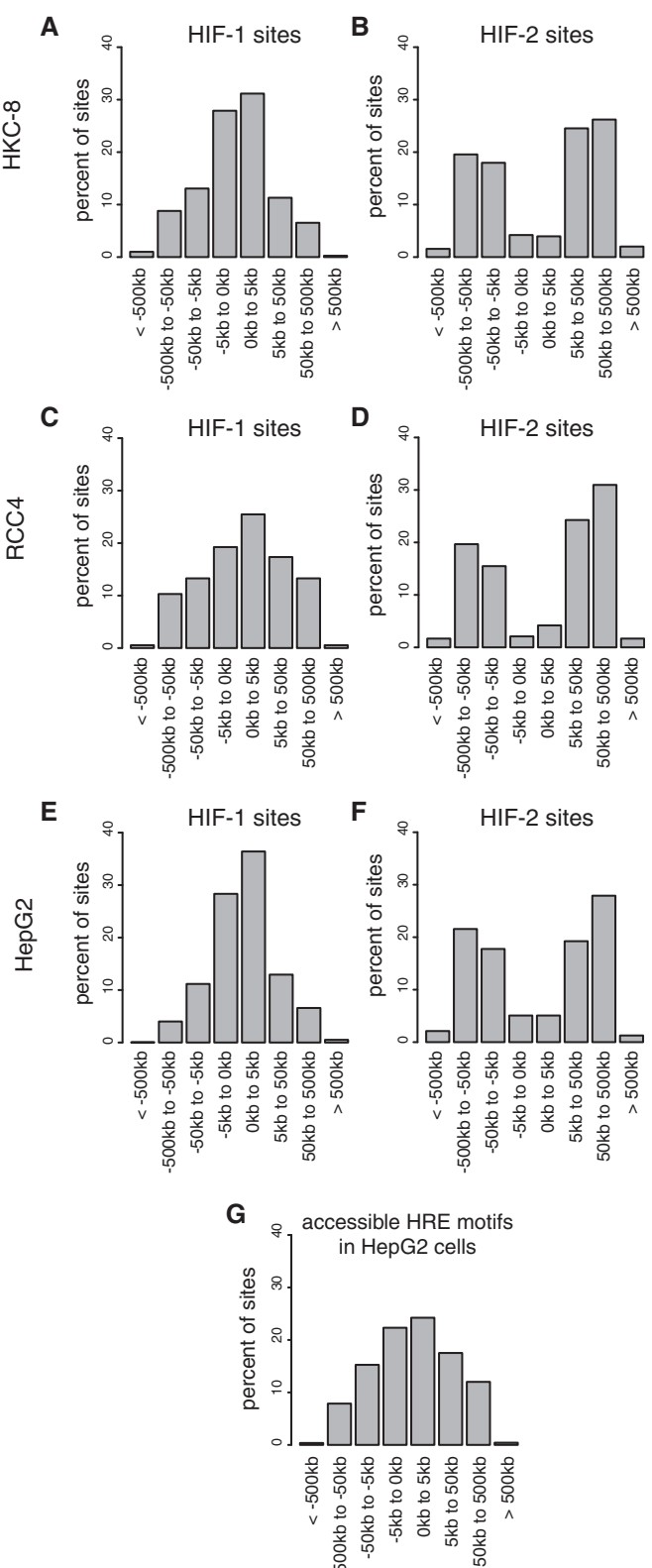

Figure 6.   **Distinct binding distributions of HIF-1 and HIF-2 are conserved across cell types.**

A–F   Canonical HIF-1 and HIF-2 binding sites were determined by overlap of HIF-α and HIF-1β binding sites identified by the MACS peak caller as for Fig 4. The distance from each binding site to the nearest annotated gene promoter was determined and plotted as a frequency distribution for (A) HIF-1 and (B) HIF-2 sites in HKC-8 cells, (C) HIF-1 and (D) HIF-2 sites in RCC4 cells, and (E) HIF-1 and (F) HIF-2 sites in HepG2 cells. The distribution of HIF-1 binding sites was consistently more promoter-proximal than HIF-2 sites despite the sites themselves differing between cell types.

G   The distribution of accessible HRE motifs was also plotted in HepG2 cells and differed significantly from that of both HIF-1 ($P\sim10^{-50}$) and HIF-2 ($P\sim10^{-80}$) binding sites (chi-squared test).

HIF-1 and HIF-2 sites are specifically enriched in each cell line for particular transcription factor binding motifs specifically (Tables EV1–EV3). Notably, in HKC-8 and RCC4 cells, HEY1/2 and ZNF263 motifs were amongst those most enriched at HIF-1 binding sites, whilst AP-1 motifs were most markedly enriched at HIF-2 binding sites. In HepG2 cells, the most enriched motifs at HIF-1 sites included SP1/2 as well as HEY2, whilst those most enriched at HIF-2 sites included FOXD2, FOXL1 and FOXC2. However, enrichment of multiple closely related motifs was frequently observed and that data do not indicate which, if any, transcription factors are actually bound.

We therefore examined canonical HIF-1 and HIF-2 binding sites in HepG2 cells for their association with the binding of other transcription factors using ENCODE ChIP-seq data for 61 additional DNA-binding proteins that had been experimentally defined in the same cell line (Fig 8B). Several DNA-binding proteins were strongly enriched at both HIF-1 and HIF-2 sites, including CEBPD, BHLHE40, MYBL2, SP1, EP300, FOSL2 and HDAC2. Consistent with their promoter-proximal binding distribution, HIF-1 sites were enriched for several members of the basal transcriptional machinery, including TAF1 and TBP and both total and phosphorylated RNApol2. HIF-1 sites rather than HIF-2 sites were also enriched for MYC and MAX as well as the MYC antagonists MXI1, REST1 and SIN3A. Notably, HIF-2 associated with both FOXA1 and FOXA2 binding sites in HepG2 cells. Originally described as transcriptional activators for a number of liver-specific transcripts, these are thought to act as pioneer factors that help establish tissue-specific gene expression and regulation in differentiated tissues. Similarly, both hepatocyte nuclear factors HNF4A and HNF4G are more strongly associated with HIF-2 sites than with HIF-1 sites. This suggests that the more tissue-specific role of HIF-2 may arise at least in part from association with other tissue-specific transcription factors.

## Discussion

Here, we present a systematic analysis of the pan-genomic distributions of the two major isoforms of HIF under different severity and duration of hypoxia, in different cell types, and following genetic ablation of one or other isoform. We chose to examine HIF binding specifically, since this is an important determinant of transcriptional induction by hypoxia that can be accurately defined across the genome using ChIP-seq. Though multiple other factors will shape

and HIF-2 binding sites is consistent with HIF-1 binding predominantly at promoter regulatory regions, whilst HIF-2 binds mainly to functional enhancers.

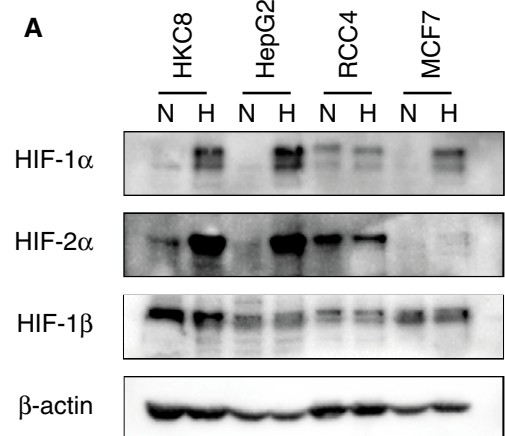

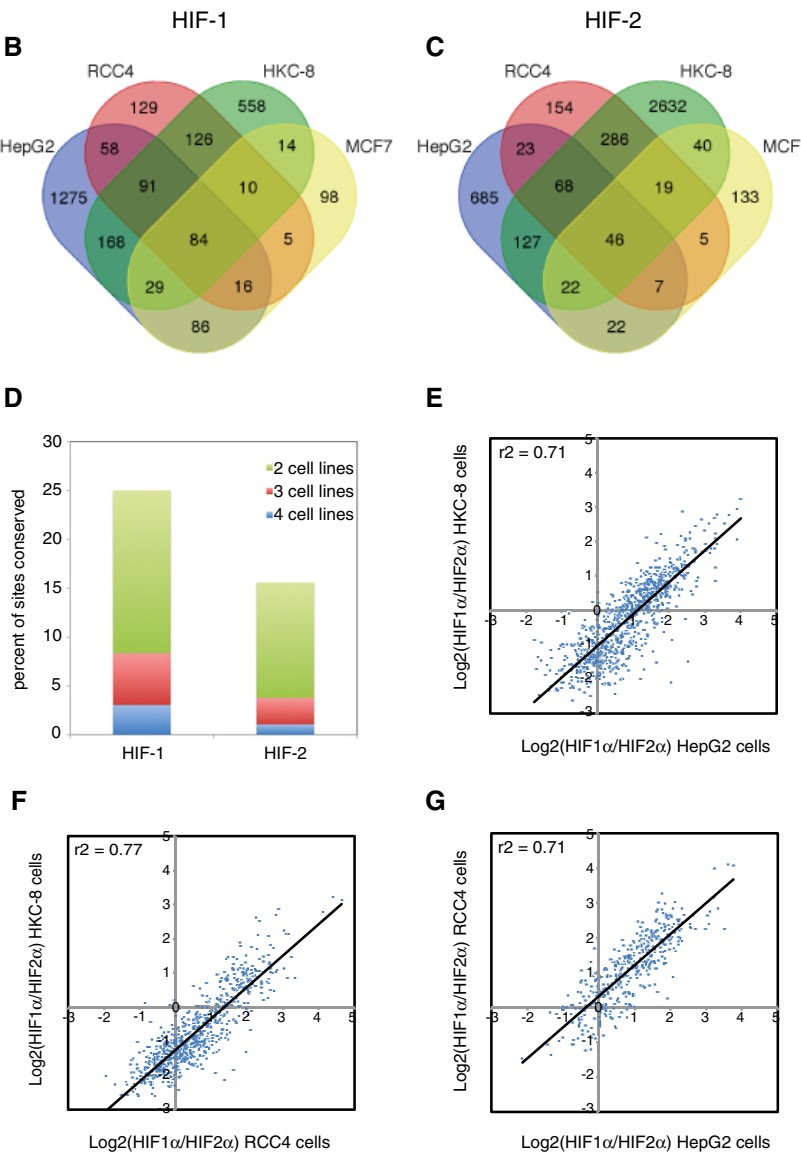

**Figure 7.**

**Figure 7.  Conservation of HIF-1α and HIF-2α specificity between cell types.**

A    Immunoblots showing HIF-1α, HIF-2α and HIF-1β protein levels in HKC-8, HepG2, RCC4 and MCF-7 cell lines in normoxia and following 16-h incubation at 0.5%
     oxygen.
B, C    Sites that bound (B) canonical HIF-1 (i.e. identified in both HIF-1α and both HIF-1β ChIP-seq experiments) or (C) canonical HIF-2 (i.e. identified in both HIF-2α and
     both HIF-1β ChIP-seq experiments) were identified from ChIP-seq datasets in HepG2 and HKC-8 cells following 16-h incubation in 0.5% oxygen and from normoxic
     RCC4 cells and compared with previously published data from MCF-7 cells.
D    The proportion of total HIF-1 or HIF-2 sites that were shared between two, three and four cell lines is plotted.
E–G    For sites that are shared between (E) HepG2 and HKC-8 cells, (F) HKC-8 and RCC4 cells, or (G) RCC4 and HepG2 cells, the ratio of HIF-1α/HIF-2α signal for one cell
     line was plotted against the other. The ratio of HIF-1α/HIF-2α in one cell line correlates well with that in the other. ChIP-seq signal is averaged across two
     independent experiments.

changes in the transcriptome in hypoxic cells, our findings provide several new insights into mechanisms that mediate different components of the HIF pathway.

Both HIF-1α and HIF-2α bound to DNA as heterodimers with HIF-1β, with no clear evidence for either non-stoichiometric binding of α/β polypeptides, or isolated "non-canonical" binding of either HIF-α polypeptide in the absence of HIF-1β. Although we have previously observed evidence for binding of HIF-α, particularly HIF-2α, to DNA in apparent excess over HIF-1β following over-expression [16], the current findings suggest that this is unusual under endogenous conditions. We cannot exclude the possibility that binding of HIF-α to DNA with other binding partners may occur, but evade capture in these assays, due to reduced affinity or cross-linking efficiency. Nevertheless, our findings are consistent with the established canonical mode of HIF binding to DNA as an α/β heterodimer being the most prevalent mode of binding.

As expected, HIF-α polypeptides were induced more strongly by more severe hypoxia [39]. Surprisingly, however, despite a very wide range in the strength of binding signals between different sites, we found little evidence that sites with stronger signals in less severe hypoxia became saturated as severity was increased and little evidence for the redistribution of binding signals to other sites. Rather, the dominant pattern was that binding at specific sites increased in proportion to the total binding across the genome, with any tendency for sites to load earlier or later in respect of the severity of hypoxia, being very modest. This behaviour was also reflected in the analyses performed after different duration of hypoxic exposure. Again, the binding for each HIF isoform at individual sites was found to be proportional to the total binding across the genome for that isoform, despite the different HIF isoforms displaying quite different responses to the duration of hypoxia, although it is possible that more extended periods of hypoxia could alter HIF binding by altering chromatin structure. These experiments suggest that despite many sites binding both HIF isoforms to a greater or lesser extent, binding distributions are determined by the intrinsic properties of each isoform, with little cross-competition, at least under the conditions of our experiments.

Strong support for intrinsically distinct patterns of binding was provided by experiments using genetically engineered cells, in which disruption of either HIF-α gene had very little effect on the binding of the other. Remarkably, this was observed across HIF binding sites displaying wide-ranging levels of specificity for one or other isoform, with little evidence for cross-compensation at sites that bound both HIF-α isoforms in wild-type cells. Overall, the findings therefore support a model in which HIF-1 and HIF-2 loading at HIF binding sites is an intrinsic property of each isoform and

broadly proportional to the availability of each isoform for binding under the particular conditions of hypoxic exposure.

Analysis of sites at which both HIF isoforms were observed to bind revealed no evidence for multiple HIF binding sites. Thus, it appears that binding must occur in a non-competitive and non-compensatory manner at the same sites. Whilst at first sight, this might appear surprising it is consistent with the common observation that suppressing one HIF isoform alone has a significant effect on gene expression, which would not be the case if one isoform were able to compensate for the other. It is also consistent with a model in which HIF binding to chromatin is very transient leading to low levels of occupancy. Indeed, HIF binding signals vary by up to 100-fold between different sites, so that even if the strongest sites are fully occupied, most sites must be occupied in only a small proportion of cells at any one time.

When patterns of HIF binding were analysed in different cell lines, a high level of cell-type specificity was observed. Thus, when categorical analyses of HIF-1 binding sites were considered, only 3% of sites exceeded computer-generated "peak calling" thresholds across all four cell lines that were tested. Interestingly, when these differences in HIF binding were interrogated quantitatively, a continuum of binding was observed across apparently selective HIF binding sites, such that most sites retained levels of HIF binding that were above background in the "non-permissive" cell types, even though they were much lower than in the "index" cell and well beneath computer-generated peak calling thresholds. Thus, cell-type binding specificity appears to reflect a continuum of binding, rather than an all or none process. Whether this reflects different distributions of binding across a population of cells, or different temporally defined binding probabilities within a population of cells, will require further analyses at the single-cell level. When HIF-1 and HIF-2 binding were compared, differences in the extent of cell-type specificity were observed, with HIF-2α being substantially more cell-type-specific. However, despite marked cell-type specificity, several distinct characteristics of HIF-1 and HIF-2 binding were preserved. First, distinct patterns were observed in relation to the distance of binding sites from promoters. In each cell type, when the distribution of HIF binding was considered in relation to the distribution of available hypoxia response elements at gene loci, a clear bias was observed for HIF-2 to bind more distantly from transcriptional start sites and for HIF-1 to bind more closely to transcriptional start sites. Second, when sites that were common across cell types were considered, the ratio of HIF-1α to HIF-2α was found to be very similar between different cell types. Taken together, these findings imply that factors determining HIF-α isoform binding specificity are distinct from those determining the cell-type specificity of HIF binding and are likely conserved between cell types. The observation

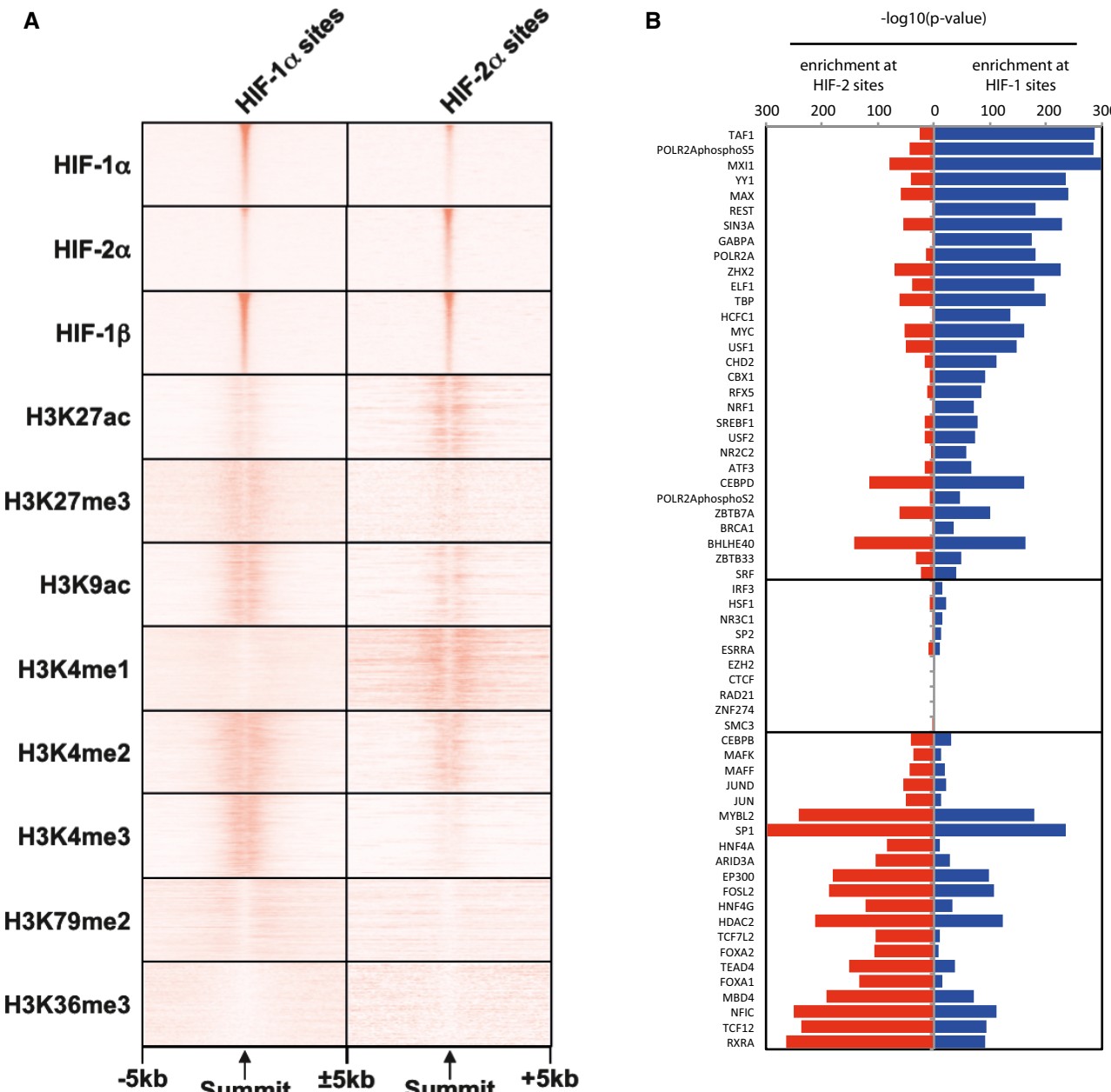

**Figure 8.  Transcriptional landscape at HIF-1 and HIF-2 binding sites in HepG2 cells.**

A  Heatmaps showing average (*n* = 2, independent ChIP-seq experiments) ChIP-seq signal (read counts per million mapped reads, CPM; expressed as colour intensity) at canonical HIF-1 and HIF-2 binding sites and across the flanking ±5 kb regions (*x*-axis) in HepG2 cells incubated in 0.5% oxygen for 16 h. Sites are ordered on the *y*-axis according to HIF signal derived from HepG2 cells incubated in hypoxia. ChIP-seq signal for histone modifications was obtained from ENCODE and was derived from HepG2 cells incubated in normoxia.

B  Bar chart showing enrichment of transcription factor binding sites (TFBSs) at canonical HIF-1 and HIF-2 sites in HepG2 cells (identified as in Fig 4). TFBSs for 61 additional DNA-binding proteins were determined from ENCODE ChIP-seq data in normoxic HepG2 cells. The significance of the overlap between each set of binding sites and canonical HIF-1 and HIF-2 sites was determined using a hypergeometric test. Accessible sites determined from ENCODE DNase-seq data in HepG2 cells were used as a negative control.

that both HIF-1 and HIF-2 binding patterns deviate (in different directions) from the distribution of available hypoxia response elements in respect of distance from promoters indicates that both complexes must respond to influences that bias their binding with respect to distance from promoters. Analysis of publically available

ENCODE ChIP-seq data in HepG2 cells revealed that other transcription factors (e.g. CEBPB and CEBPD) may behave similarly, with closely related isoforms demonstrating distinct distributions with respect to distance from promoters. Nevertheless, to our knowledge, this behaviour remains largely unexplained. In other work, we have

demonstrated physical contact between HIF binding sites and hypoxia responsive promoters lying at larger distances [40], suggesting that many of these distant sites are enhancers. In keeping with this, we observed differential associations between HIF-1 and HIF-2 binding sites with histone modifications that are characteristic of promoters and enhancers, respectively [36]. We also observed marked differences in available transcription factor binding sites and in the binding of other transcription factors within the vicinity of HIF-1 versus HIF-2, suggesting that specific interactions with other transcription factors might mediate the binding specificity of HIF isoforms. However, these associations do not necessarily imply causation, and the mechanisms distinguishing promoter and enhancer binding and other aspects of HIF isoform selectivity will require further investigation.

Overall, the work revealed surprisingly discrete binding patterns for HIF-1 and HIF-2. They suggest that the DNA-binding complexes function more independently than previously foreseen and rationalize the use of isoform-specific therapeutic inhibitors for specific indications.

# Materials and Methods

## Cell culture

HKC-8 cells were a gift from L.C. Racusen [41]. RCC4 cells were a gift from C.H. Buys. The identity of RCC4 was confirmed through the presence in RNA-seq datasets of unique mutations in the coding region of the *VHL* gene (chr3:10,183,841 G>del) that are as previously described. HepG2 cells were purchased directly from ATCC and validated by STR genotyping. Cell lines were grown in Dulbecco's modified Eagle's medium, 100 U/ml penicillin, 100 μg/ml streptomycin and 10% foetal bovine serum (Sigma-Aldrich) and regularly tested for mycoplasma infection. Hypoxic incubations were performed for the specified duration and ambient oxygen concentration in an In Vivo2 400 Hypoxia Workstation (Ruskinn Technology).

## Immunoblot analysis

Cells were lysed in NP-40 buffer, and proteins were resolved by SDS–PAGE. After transferring the proteins onto PVDF membranes, HIF proteins were detected using anti-HIF-1α (mouse monoclonal, BD Bioscience 610958), anti-HIF-2α (mouse monoclonal, 190b) or anti-HIF-1β (rabbit polyclonal, Novus Biologicals, NB100-110) antibodies and horseradish peroxidase-conjugated anti-mouse or anti-rabbit secondary antibodies (Dako). HRP-conjugated anti-β-actin antibody (Abcam) was used as a loading control.

## CRISPR-Cas9 disruption of HIF-1α and HIF-2α expression

Guide RNAs were designed using the CRISPR design tool (http://crispr.mit.edu/) [42]. HIF-1α and HIF-2α were targeted using the following pairs of guide RNAs: TGTGAGTTCGCATCTTGATA and GAAGGTGTATTACACTCAAG, targeting exon 2 of HIF-1α; and GCAGATGGACAACTTGTACC and TTGGAGGGTTTCATTGCCG, targeting exon 3 of HIF-2α. pSpCas9n(BB)-2A-GFP (PX461) was a gift from Feng Zhang (Addgene plasmid # 48140) [43].

## Chromatin immunoprecipitation

Chromatin immunoprecipitation (ChIP) experiments were performed as previously described [7,43–45] using antibodies directed against HIF-1α (rabbit polyclonal, PM14), HIF-2α (rabbit polyclonal, PM9) or HIF-1β (rabbit polyclonal, Novus Biologicals, NB100-110). All ChIP-seq experiments were performed in duplicate in accordance with ENCODE consortium guidelines (https://www.encodeproject.org/documents/ceb172ef-7474-4cd6-bfd2-5e8e6e38592e/@@download/attachment/ChIP-seq_ENCODE3_v3.0.pdf).

## PolyA+ selected RNA-seq

Total RNA was prepared using the mirVana miRNA Isolation Kit (Ambion, Life Technologies Ltd, Paisley, UK) and treated with DNaseI (TURBO DNA-free, Ambion). PolyA+ RNA libraries were then prepared using the ScriptSeq v2 RNA-seq Kit (Epicentre, Madison, WI, USA). All RNA-seq experiments were performed in triplicate in accordance with ENCODE consortium guidelines (https://www.encodeproject.org/documents/cede0cbe-d324-4ce7-ace4-f0c3eddf5972/@@download/attachment/ENCODE%20Best%20Practices%20for%20RNA_v2.pdf).

## High-throughput sequencing

All sequencing was performed on the HiSeq 2500 or HiSeq 4000 platforms according to Illumina protocols (Illumina, San Diego, CA, USA).

## Accession codes

ChIP-seq and RNA-seq data are available from Gene Expression Omnibus (GSE120885, GSE120886 and GSE120887).

## Bioinformatic analysis of ChIP-seq data

### Preliminary analysis
Illumina adaptor sequences were trimmed using TrimGalore (0.3.3), and reads were aligned to Genome Reference Consortium GRCh37 (hg19) using BWA (0.7.5a-r405). Low-quality mapping was removed (MapQ < 15) using SAMtools (0.1.19) [44] and reads mapping to Duke Encode black list regions (http://hgwdev.cse.ucsc.edu/cgi-bin/hgFileUi?db = hg19&g = wgEncodeMapability) were excluded using BEDTools (2.17.0) [45]. Duplicate reads were marked for exclusion using Picard tools (1.106) (http://broadinstitute.github.io/picard/). Read densities were normalized and expressed as reads per kilobase per million reads (RPKM) [46]. One million random non-overlapping regions selected from ENCODE DNase Cluster II peaks (http://hgdownload.cse.ucsc.edu/goldenPath/hg19/encodeDCC/wgEncodeRegDnaseClustered/) were used as a control.

### Peak calling
ChIP-seq peaks were identified using T-PIC (Tree shape Peak Identification for ChIP-Seq) [35] and MACS (model-based analysis of ChIP-seq) [34] in control mode. Peaks detected by both peak callers were filtered quantitatively using the total count under the peak to include only peaks that were above the 99.99th percentile of random

background regions selected from the ENCODE DNase II cluster (*P*-value < 0.0001). Only peaks from each independent replicate that overlapped by at least 1 base pair (BEDTools v2.17.0 [45]) were considered.

### Quantitation of ChIP-seq signal

Manipulation of .bam files was performed using SAMtools (v0.1.19) [44]. Briefly, SAMtools merge was used to merge sorted alignment .bam files from each replicate into a single .bam output file, which was then indexed using SAMtools index. SAMtools bedcov was then used to determine the read depth per bed region, which was normalized to the total reads in each dataset determined using SAMtools flagstat.

Binding site heatmaps were generated using Ngsplot (2.08) [47]. Boxplots were generated in R using the boxplot function in the BiocGenerics (v0.20.0) package. The distance from each HIF binding site to the nearest TSS in the Ensembl hg19 database was determined using PeakAnnotator (v1.4, https://www.ebi.ac.uk/research/bertone/software#peakannotator) and plotted as a histogram in R.

### Bioinformatic analysis of RNA-seq data

#### Preliminary analysis

Adapter sequences were trimmed as above. Reads were then aligned to GRCh37 using Tophat 2.0.8b (http://ccb.jhu.edu/software/tophat/index.shtml) and bowtie 1.0.0 (http://bowtie-bio.sourceforge.net/index.shtml) and non-uniquely mapping fragments excluded using SAMtools (0.1.19) [44]. Total read counts for each UCSC-defined gene were extracted using HTSeq (0.5.4p3) [48] with "intersection-strict" mode, and significantly regulated genes were identified using DESeq2 (ref. [49]).

#### Gene set enrichment analysis (GSEA)

Gene set enrichment analysis used 10,000 permutations, weighted enrichment score and pre-ranking of genes [50]. Both differential expression significances according to DESeq2 and fold-difference between the two conditions were used to rank genes according to the equation [51].

$$\pi_i = \varphi_i(-\log_{10} Pv_i),$$

where $\varphi_i$ is the log2 fold-change, and $Pv_i$ is the *P*-value for gene *i*.

**Expanded View** for this article is available online.

## Acknowledgements

This study was funded by Cancer Research UK (DRM; A416016), the National Institute for Health Research (DRM; NIHR-RP-2016-06-004), the Deanship of Scientific Research, King Abdulaziz University, Ministry of Higher Education for Saudi Arabia (HC, DRM and PJR), the Ludwig Institute for Cancer Research (PJR) and the Wellcome Trust (PJR; 088182/Z/09/Z, 078333/Z/05/Z, WT091857MA). This work was supported by the Francis Crick Institute which receives its core funding from Cancer Research UK (FC001501), the UK Medical Research Council (FC001501), and the Wellcome Trust (FC001501). The funders had no role in study design, data collection and analysis, decision to publish, or preparation of the manuscript. We thank the Oxford Genomics Centre at the Wellcome Centre for Human Genetics (funded by Wellcome Trust grant reference 203141/Z/16/Z) for the generation and initial processing of the sequencing data.

## Author contributions

Conceptualization: PJR, DRM; Design: all authors; Acquisition: JAS, MS, NM, PDS, EM, VN, MEC; Analysis: JAS, MS, RS, DRM; Interpretation: JAS, MS, NM, RS, HC, PJR, DRM; Original draft: JAS, MS, NM, DRM, PJR; Revision, editing and final approval: all authors.

## Conflict of interest

Peter J Ratcliffe is a scientific co-founder of ReOx Ltd., a company, which is developing inhibitors of the HIF hydroxylase enzymes. The other authors declare that they have no conflict of interest.

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
