## [Review Process File · EMBO Reports]

Inherent DNA binding specificities of the HIF-1a and HIF-2a transcription factors in chromatin

James A Smythies, Min Sun, Norma Masson, Rafik Salama, Peter D Simpson, Elizabeth Murray, Viviana Neumann, Matthew E Cockman, Hani Choudhry, Peter J Ratcliffe, David R Mole

Review timeline:	Submission date:	10 May 2018
	Editorial Decision:	18 June 2018
	Revision received:	15 August 2018
	Editorial Decision:	24 September 2018
	Revision received:	9 October 2018
	Accepted:	17 October 2018

Editor: Esther Schnapp

Transaction Report:

1st Editorial Decision

18 June 2018

Thank you for your patience while your manuscript was peer-reviewed at EMBO reports. We have now received the referee comments as well as cross-comments that are all pasted below.

As you will see, all referees acknowledge that the findings are interesting. While referees 2 and 3 are overall satisfied with your study, referee 1 points out that functional analyses would need to be provided, and that the most important conclusions would need to be confirmed with an independent cell line.

However, the referee cross-comments show that referees 2 and 3 do not feel that novel transcriptomics data and independent cell lines are essential for the publication of this study here. I therefore suggest that you address all referee comments to the best of your abilities and may be provide some more data on HIFa target gene expression in different cell lines, as suggested by the referees. We could also talk about the experiments for the revisions on the phone, if you like.

Given the constructive comments, I would thus like to invite you to revise your manuscript with the understanding that the referee concerns must be fully addressed and their suggestions taken on board. Please address all referee concerns in a complete point-by-point response. Acceptance of the manuscript will depend on a positive outcome of a second round of review. It is EMBO reports policy to allow a single round of revision only and acceptance or rejection of the manuscript will therefore depend on the completeness of your responses included in the next, final version of the manuscript.

Revised manuscripts should be submitted within three months of a request for revision; they will otherwise be treated as new submissions. Please contact us if a 3-months time frame is not sufficient for the revisions so that we can discuss this further. Given your 6 main figures, I suggest that you layout your manuscript as a full article with separate results and discussion sections.

Regarding data quantification, please specify the number "n" for how many independent experiments were performed, the bars and error bars (e.g. SEM, SD) and the test used to calculate p-values in the respective figure legends. This information is currently incomplete and must be

provided in the figure legends.

We now strongly encourage the publication of original source data with the aim of making primary data more accessible and transparent to the reader. The source data will be published in a separate source data file online along with the accepted manuscript and will be linked to the relevant figure. If you would like to use this opportunity, please submit the source data (for example scans of entire gels or blots, data points of graphs in an excel sheet, additional images, etc.) of your key experiments together with the revised manuscript. Please include size markers for scans of entire gels, label the scans with figure and panel number, and send one PDF file per figure.

- a complete author checklist, which you can download from our author guidelines (<http://embor.embopress.org/authorguide#revision>). Please insert page numbers in the checklist to indicate where in the manuscript the requested information can be found. The completed author checklist will also be part of the RPF (see below).
- a letter detailing your responses to the referee comments in Word format (.doc)
- a Microsoft Word file (.doc) of the revised manuscript text
- editable TIFF or EPS-formatted figure files in high resolution. In order to avoid delays later in the process, please read our figure guidelines before preparing your manuscript figures at: http://www.embopress.org/sites/default/files/EMBOPress_Figure_Guidelines_061115.pdf

I look forward to seeing a revised version of your manuscript when it is ready. Please let me know if you have questions or comments regarding the revision.

REFEREE REPORTS

Referee #1:

In this paper, Smythies et al. show that pan-genomic patterns of HIF-1 α and HIF-2 α binding are largely unaffected by the extent or duration of hypoxia or the presence of the other HIF α isoform. They also demonstrate that distinct binding distribution of HIF-1 α and HIF-2 α are conserved between cell types. This is an interesting study and important for the field. Overall the scientific approaches and data presentation appear to be solid, and the text is written clearly. However, characterization of DNA binding patterns of HIFs at different biological contexts is not supported by any sort of functional analysis. This is a major weakness of the paper and needs to be addressed by global transcriptome analysis (or at least, by analysis of HIF target genes). Below are additional comments to improve the manuscript.

- (1) There is no explanation why a specific cell line, HKC-8, has been used throughout the study.
- (2) An important finding of this study is that HIF α isoform binding is mostly independent of the severity or duration of hypoxia. However, the experiments supporting this conclusion were carried out using only one cell line (Fig 2 and EV Fig3). Also, the hypoxia time points used in this study (6h, 16h, or 48h) do not cover long-term adaptation to hypoxia. Based on the biological conditions and controls tested in this study, their conclusion is too strong and needs to be toned down.

(3) Another major conclusion of the work is that two HIF α isoforms do not compete for binding site (Fig 3). This is an unexpected finding; therefore, requires rigorous experimental verification. Again, using one cell line would not be sufficient to make a firm conclusion in this case.

(4) When comparing HIF binding patterns in different cell lines (Fig 4), the authors need to provide the protein expression levels of HIF-1 α , HIF-2 α , and HIF-1 β in these cell lines at the specific hypoxia condition used for their ChIP-seq experiments.

(5) Throughout the paper, the authors should clarify the extent (O₂ level) and duration of hypoxia for each experiment. For example, the hypoxia conditions for four different cell lines (Fig 4) were not provided in the manuscript. In RCC4 cells, HIF α protein levels are high even under normoxia due to VHL deficiency. Did the authors perform ChIP-seq experiments with RCC4 cells cultured in normoxic and hypoxic conditions, and compare the differences in HIF binding patterns between normoxia and hypoxia?

Referee #2:

This report examines chromatin binding characteristics during hypoxia of HIF-1 and HIF-2, two members of the three-member HIF family that control the majority of HIF signaling in many mammalian cells. This is a superbly executed study that clearly and cleanly documents the dynamics of HIF-1 and HIF-2 recruitment under severe and modest hypoxic conditions. Using an experimental approach that emphasizes precision, the authors provide several lines of evidence for important differences between HIF-1 and HIF-2 binding in hypoxia "permissive" and "non-permissive" cell lines. These differences are consistent with other specific biological roles observed for these factors in the intact mammal. The data is convincing and includes appropriate controls. The replication of all experiments in each respective cell line provides a thorough palette for comparison.

My only suggestion would be to speculate more in the discussion about the differences between HIF-1 and HIF-2 observed in this study. Specifically, why HIF-2 apparently has reduced binding in the proximal promoter (-5kb to 0kb) and coding region of genes (0kb to 5kb) in contrast to HIF-1, which appears to have peak binding in these same regions. Are these difference in recruitment somehow linked to the differences in associated chromatin marks? The authors correctly point out that association does not equate with causality. However, it seems reasonable to assume that there may be a connection. Are there data from other studies that indicate HIF-1 and/or HIF-2 selectively interact with cellular factors that generate these marks?

Referee #3:

In this manuscript, Smythies et al. have investigated the interesting topic of what governs DNA binding specificity of the HIF1 α and HIF2 α transcription factors. While both alpha subunits (HIF1 α /ARNT and HIF2 α /ARNT heterodimers) bind to identical consensus sequences, their genome-wide binding profiles and target genes are only partially overlapping. The determinants of their binding specificity and whether it relates to severity/duration of hypoxia, cell type or competition between both isoforms is unknown. To answer these long-standing questions, the authors used ChIP-seq to analyze the binding distribution of HIF subunits in response to varying hypoxic conditions in multiple cell types. In addition, they studied the effect of deletion of one HIF α isoform on the pan-genomic distribution of the other. Altogether the presented data strongly support that HIF- α subunits bind DNA with distinct and characteristic distribution patterns that are largely unaffected by the intensity or duration of the hypoxic stimuli. The data also shows that, both HIF- α subunits bind chromatin in a stoichiometric ratio with HIF- β and that their binding pattern is independent on the presence/absence of the other alpha subunit. In summary, the manuscript report that the pan-genomic distribution of HIF is an inherent property of each alpha subunit that is largely independent of the hypoxic stimulus or competition between isoforms. In this regard, it is a significant work that provides novel insight into a long-standing question in the field. In addition, the conclusions could be of interest to a wide range of researchers as they are relevant to understand the general mechanisms governing gene expression regulation and how to exploit them to selectively target pathways relevant to human disease. Finally, the main

findings reported in the manuscript are robustly supported by the results and are based on state-of-the-art experimental approaches.

Specific queries:

1. Authors convincingly demonstrate that HIFbeta is universally present at HIFalpha binding sites with no evidence of HIFalpha binding in the absence of HIFbeta. Is the reverse true? HIFbeta is known to have dimerisation partners other than HIFalpha, thus it would be interesting to analyze the possibility of HIFbeta binding in the absence of HIFalpha in the cell lines included in the study. For example, the authors could show the combined HIFalpha signal intensity at all HIFbeta binding sites and compare with the mean HIFalpha intensity across non-HIFbeta bound enhancers/promoters. In addition, does HIFbeta bind DNA in normoxia?
2. Data presented shows that although the majority of HIF binding sites loaded similarly as the severity of hypoxia increases, there are a limited number of "progressive loading" and "early loading" sites. Could these sites represent low and high affinity binding sites respectively? Do these types of sites show different nucleotide composition (e.g. different frequency of A/G at the first position of the RCGTG motif or in the nucleotides flanking this core). Additionally, the representation of the ratio of HIFalpha ChIP-seq signal at 0.5% compared to 3% against the total (0.5%+3%) could reveal a relationship between total binding (as readout of binding affinity) and loading.
3. A key and important finding of this work is that HIF1 and HIF2 loading at HIF binding sites is an intrinsic property of each isoform with each isoform showing a clear and strong bias regarding their location relative to the TSS. However, the determinants driving this selectivity are unclear. The authors show differential associations of HIF1 and HIF2 binding sites with histone modifications but these are very general marks and, as stated in the discussion, it is difficult to determine causality. Thus, it is likely that other factors determine HIF binding specificity. In this regard, is there any sequence motif significantly over-represented (depleted) in the HIF2 "only" binding sites compared with HIF1a "only"?
4. The data presented demonstrate that many sites are bound by both isoforms yet there is little or no cross-competition, suggesting that common sites could be co-occupied rather than simultaneously bound by one or the other isoform. Is there evidence supporting or ruling out this possibility? Do shared sites have a larger number of RCGTG motifs on average? How is the distribution of distances between HIF1a and HIF2a peak summits at shared sites? How are HIFbeta peaks at these sites compared to those at sites bound by a single alpha isoform?
5. CRISPR-mediated ablation of HIF2a results in a slight reduction of HIF1a protein. Assuming that these are clonal cell lines (it is not explicitly indicated in the manuscript), is this an anecdotal effect due to cell-to-cell variability or is it a reproducible effect seen in other cell clones? Does the ablation of one isoform affect the mRNA levels of the other?
6. Regarding conservation of HIF binding sites across cell lines, it is clear that categorical classification based on peak calling underestimates overlap. The inclusion of a base-line, such as that shown in figures 1E and 1F, in the quantitative analysis shown in expanded figures 4G and 4H, could be helpful to get a better approximation to the number of overlapping sites across cell lines. From a functional perspective, it would be nice to compare the overlap between the genes nearest to HIF-bound sites across cell lines.

Cross-comments from referee 2:

Both of these suggestions in theory would expand the relevance of this study. However, in reality, they potentially pose significant theoretical and practical barriers. Gene expression analyses might indicate what HIF binding sites are transcriptionally active. However, given that HIF binding appears to affect epigenetic signatures that may also be influenced in a gene-specific manner by non-HIF factors, one could imagine an experimental result that provides lots of data, but one without any meaningful pattern, at least at this time. Nevertheless, it may be reasonable to suggest or request that the authors provide either broad transcriptional analyses or targeted rtPCR analyses in the three cell lines examined (HKC-8, RCC4, HepG2) using RNA/cDNA samples that were prepared in parallel.

For targeted rtPCR analyses, it would be helpful to examine a dozen or so select genes that are similar as well as ones that differ between the three cell lines.

However, my main concern with asking for additional transcriptional, and especially more cellular, studies is the practical barrier that this would impose on the authors. ChIPseq experiments are not trivial ones in terms of time, money, and effort. Asking for additional cell line studies, in particular, would likely be a death knell for this study and would reinforce an unfortunate pattern of electronic "piling-on" that permeates the review process these days. This study has enough merit with its current content to warrant publication now, in my opinion.

Cross-comments from referee 3:

Yes, I agree. Global transcriptome analyses of HIF target gene expression could certainly aid to determine the functional impact of the of different DNA binding patterns of HIFs in different biological contexts. On the other hand, the confirmation of results in a second line will strengthen the conclusions.

However, asking for the generation of additional CRISPR-edited cell lines implies an awful amount of work just to get a confirmatory result rather than providing novel insights. Thus, in my opinion, other (independent) works should confirm the lack of competition between both HIF isoforms. As regards of transcriptome analysis, authors could make use of published RNA-seq datasets to correlate expression with binding patterns"

Thus, as indicated in my evaluation report and in agreement with referee #2, I believe the work has sufficient merit to be published after minor modifications addressing the points I raised before.

1st Revision - authors' response

15 August 2018

Response to reviewers

Referee #1:

In this paper, Smythies et al. show that pan-genomic patterns of HIF-1 α and HIF-2 α binding are largely unaffected by the extent or duration of hypoxia or the presence of the other HIF α isoform. They also demonstrate that distinct binding distribution of HIF- 1 α and HIF-2 α are conserved between cell types. This is an interesting study and important for the field. Overall the scientific approaches and data presentation appear to be solid, and the text is written clearly. However, characterization of DNA binding patterns of HIFs at different biological contexts is not supported by any sort of functional analysis. This is a major weakness of the paper and needs to be addressed by global transcriptome analysis (or at least, by analysis of HIF target genes).

Thank you. We understand that functional analyses (we assume of hypoxia-inducible gene expression) are of interest and indeed many such studies have been reported. However, attempting an accurate, mechanistically meaningful correlation of the current data with transcriptomic data is much less straightforward than might appear.

First – differences in time course (between binding and transcriptional target abundance) are problematic, since neither will come into steady state – at least not at the same time.

Second – very many factors influence transcript abundance other than DNA binding of a relevant transcription factor. For example:

- **FIH regulates the interaction of HIF-1 α with the transcriptional co-activator p300/CBP and operates over a different oxygen range to the PHD enzymes that regulate HIF abundance, and hence HIF binding.**
- **Some genes are differentially regulated by HIF-1 or HIF-2, even when both isoforms are bound**
- **Transcript abundance will depend upon the duration of transcriptional activation as well as the balance between transcript generation and degradation rates.**

- Expression levels will depend on interaction with other transcription factors that are both general and cell-type specific.

•
 These factors will greatly complicate global transcriptome analysis and a proper interpretation will require not only the measurement of nascent transcript levels, but also of the binding of RNAPol2, transcriptional co-activators and other as yet unidentified transcription factors. As indicated by referees 2 and 3 in the cross comments, meaningful mechanistic analyses would be complex and beyond the scope of the current manuscript.

In this work, we chose to analyse, in depth, the factors constraining HIF isoform-specific binding, since this is a definable property whose description should open a clear route to further mechanistic analysis. Much previous literature has demonstrated that DNA binding of this (and other) transcription factors alters gene expression so we are confident of the biological relevance of our work.

However, considering this request, the suggestions made by referee 3 and the cross comments by referee 2, we have provided broad correlative RNA-seq transcriptome data for the three cell lines HKC8, RCC4 and HepG2 in normoxia and following 16 hours incubation in 0.5% hypoxia (we have previously published this analysis for MCF-7 cells in Schodel et al, *Blood* 2011). This demonstrates the expected overall association between HIF-binding and hypoxic gene expression using Gene Set Enrichment Analysis. We have added this analysis as an Expanded View Figure and briefly discussed some of the points outlined above in the revision.

Below are additional comments to improve the manuscript.

(1) There is no explanation why a specific cell line, HKC-8, has been used throughout the study. **Our laboratory is interested in the key role of the HIF pathway in the pathogenesis of clear cell kidney cancer, and we have studied this immortalized cell line derived from proximal renal tubules cells extensively, as a background to that work. More importantly for the current work, we have used the same cell line throughout to enable comparison of data sets. In addition, HKC8 cells express both HIF- α isoforms at reasonable (though not exceptionally high) levels and so provide a good basis for the current studies.**

(2) An important finding of this study is that HIF α isoform binding is mostly independent of the severity or duration of hypoxia. However, the experiments supporting this conclusion were carried out using only one cell line (Fig 2 and EV Fig3). Also, the hypoxia time points used in this study (6h, 16h, or 48h) do not cover long-term adaptation to hypoxia. Based on the biological conditions and controls tested in this study, their conclusion is too strong and needs to be toned down. **We understand the concerns about the generality of observations arising from experiments in one cell line and have added a caveat to this effect in the revised manuscript. However, we feel that the ability to correlate multiple datasets within one tightly defined setting provides a powerful tool in dissecting the HIF response and have chosen to focus our resources in this way.**

As pointed out in the cross-comments, repeating the experimental plan at this level of detail in a different cell is a huge piece of work beyond our resources and we agree that it would be of greater value to the scientific community to await other independent work post-publication. We also agree that measuring the responses to long-term hypoxia could be of interest in future studies. However mechanistic understanding of long-term adaptation to hypoxia is likely to be highly complex. One particular issue we think is likely to be very important in cell culture is the potential to select variants that are present in all tissue culture cell populations when stresses are applied for periods that are long in relation to cell cycle times. We are currently considering how best to deploy lineage marking and single cell methodologies that should address this problem, but that work is outside the scope of this paper.

We therefore focused our analysis within the timeframe and severity of hypoxia that is known to fully induce the HIF response in a manner that is largely unconfounded by these effects. We have briefly discussed these issues in the revised manuscript, including moderation of the conclusions, as suggested by the reviewer.

(3) Another major conclusion of the work is that two HIF α isoforms do not compete for binding site (Fig 3). This is an unexpected finding; therefore, requires rigorous experimental verification. Again, using one cell line would not be sufficient to make a firm conclusion in this case.

We agree that the lack of competition between the two HIF isoforms is a novel finding that might initially appear unexpected. However, it is consistent with other observations. Firstly, HIF-binding signals vary by up to 100-fold between different sites (e.g. Figure 1, Expanded View Figures 1 and 2). Thus even if the strongest sites are 100% occupied, the vast majority of sites will have low levels of occupancy and we believe that this is the reason why the two isoforms do not compete for DNA binding (i.e. binding is very transient, so that binding of one isoform does not affect the kinetic of binding to the other isoform). Secondly, our finding is consistent with the common observation that suppressing one HIF-isoform alone has a significant effect on gene expression. If one isoform were able to compensate for the other then suppressing each individually would have little effect.

We have added a brief discussion of these points in the revised manuscript.

(4) When comparing HIF binding patterns in different cell lines (Fig 4), the authors need to provide the protein expression levels of HIF-1 α , HIF-2 α , and HIF-1 β in these cell lines at the specific hypoxia condition used for their ChIP-seq experiments.

We have now included these immunoblots as Panel A in Figure 4.

(5) Throughout the paper, the authors should clarify the extent (O₂ level) and duration of hypoxia for each experiment. For example, the hypoxia conditions for four different cell lines (Fig 4) were not provided in the manuscript. In RCC4 cells, HIF α protein levels are high even under normoxia due to VHL deficiency. Did the authors perform ChIP-seq experiments with RCC4 cells cultured in normoxic and hypoxic conditions, and compare the differences in HIF binding patterns between normoxia and hypoxia?

We apologize for this omission and have amended the manuscript. As the referee points out, in RCC4 cells, HIF- α levels are constitutively high. In these cells, experiments were performed in normoxic conditions. We did not compare binding patterns in normoxic and hypoxia RCC4 cells, though in other work we have done this for the pVHL-defective cell line 786-0 and observed very few differences.

Referee #2:

This report examines chromatin binding characteristics during hypoxia of HIF-1 and HIF-2, two members of the three-member HIF family that control the majority of HIF signaling in many mammalian cells. This is a superbly executed study that clearly and cleanly documents the dynamics of HIF-1 and HIF-2 recruitment under severe and modest hypoxic conditions. Using an experimental approach that emphasizes precision, the authors provide several lines of evidence for important differences between HIF-1 and HIF-2 binding in hypoxia "permissive" and "non-permissive" cell lines. These differences are consistent with other specific biological roles observed for these factors in the intact mammal. The data is convincing and includes appropriate controls. The replication of all experiments in each respective cell line provides a thorough palette for comparison.

My only suggestion would be to speculate more in the discussion about the differences between HIF-1 and HIF-2 observed in this study. Specifically, why HIF-2 apparently has reduced binding in the proximal promoter (-5kb to 0kb) and coding region of genes (0kb to 5kb) in contrast to HIF-1, which appears to have peak binding in these same regions. Are these difference in recruitment somehow linked to the differences in associated chromatin marks? The authors correctly point out that association does not equate with causality. However, it seems reasonable to assume that there may be a connection. Are there data from other studies that indicate HIF-1 and/or HIF-2 selectively interact with cellular factors that generate these marks?

Thank you. We know of no protein-protein interaction data (e.g. immunoprecipitation followed by mass-spectrometry) that has consistently identified DNA-binding proteins that specifically interact with one HIF-isoform, though we agree that this is an important question. Furthermore, since HIF binding to DNA is transient and the number of HIF-molecules bound to DNA in any one cell at any one time is small (we estimate at most a few hundred molecules),

such interactions will likely occur in a very low stoichiometric ratio and be challenging to define.

However, we have correlated our HIF-1 and HIF-2 binding datasets in HepG2 cells with publically available transcription factor ChIP-seq datasets in the same cell line. This has identified non-HIF transcription factors and DNA-binding proteins that are present selectively at HIF-1 or HIF-2 sites and we have added this analysis to Figure 6 of the manuscript. We again emphasize that such associations do not equate with causality. Experiments to distinguish between these possibilities are beyond the scope of the current manuscript.

Referee #3:

In this manuscript, Smythies et al. have investigated the interesting topic of what governs DNA binding specificity of the HIF1a and HIF2a transcription factors. While both alpha subunits (HIF1a/ARNT and HIF2a/ARNT heterodimers) bind to identical consensus sequences, their genome-wide binding profiles and target genes are only partially overlapping. The determinants of their binding specificity and whether it relates to severity/duration of hypoxia, cell type or competition between both isoforms is unknown. To answer these long-standing questions, the authors used ChIP-seq to analyze the binding distribution of HIF subunits in response to varying hypoxic conditions in multiple cell types. In addition, they studied the effect of deletion of one HIFalpha isoform on the pan-genomic distribution of the other. Altogether the presented data strongly support that HIF-alpha subunits bind DNA with distinct and characteristic distribution patterns that are largely unaffected by the intensity or duration of the hypoxic stimuli. The data also shows that, both HIF-alpha subunits bind chromatin in a stoichiometric ratio with HIF-beta and that their binding pattern is independent on the presence/absence of the other alpha subunit.

In summary, the manuscript report that the pan-genomic distribution of HIF is an inherent property of each alpha subunit that is largely independent of the hypoxic stimulus or competition between isoforms. In this regard, it is a significant work that provides novel insight into a long-standing question in the field. In addition, the conclusions could be of interest to a wide range of researchers as they are relevant to understand the general mechanisms governing gene expression regulation and how to exploit them to selectively target pathways relevant to human disease. Finally, the main findings reported in the manuscript are robustly supported by the results and are based on state-of-the-art experimental approaches.

Specific queries:

1. Authors convincingly demonstrate that HIFbeta is universally present at HIFalpha binding sites with no evidence of HIFalpha binding in the absence of HIFbeta. Is the reverse true? HIFbeta is known to have dimerisation partners other than HIFalpha, thus it would be interesting to analyze the possibility of HIFbeta binding in the absence of HIFalpha in the cell lines included in the study. For example, the authors could show the combined HIFalpha signal intensity at all HIFbeta binding sites and compare with the mean HIFalpha intensity across non-HIFbeta bound enhancers/promoters. In addition, does HIFbeta bind DNA in normoxia?

We agree that this is an interesting question. As suggested, we have repeated the reciprocal analysis to that shown in Figure 1 to look at the combined HIF- α signal at HIF-1 β binding sites ranked according to HIF- α signal and have added this data to the Expanded View Figures. Under the hypoxic culture conditions used in our experiments, this does not reveal any HIF-1 β sites that lack HIF- α signal. However, as a further check, we have also performed HIF-1 β ChIP-seq analysis in normoxic HKC8 cells in which both HIF-1 α and HIF-2 α had also been ablated by CRISPR-Cas9 induced frameshift mutations. In this setting, a much smaller number (622 versus 5177) of HIF-1 β peaks were detected and were weakly enriched for the ARNT binding motif. Interestingly, the majority (417/622) were not identified in the hypoxic wild-type cells suggesting that HIF-1 β may re-distribute to these sites in the absence of HIF- α subunits.

We have commented on this interesting point in revision, and have added some of this data. However, it would also be important to study conditions of AHR activation (one of the

alternative dimerization partners of HIF-1b), and we have made this clear in our commentary on the additional data.

- Data presented shows that although the majority of HIF binding sites loaded similarly as the severity of hypoxia increases, there are a limited number of "progressive loading" and "early loading" sites. Could these sites represent low and high affinity binding sites respectively? Do these type of sites show different nucleotide composition (e.g. different frequency of A/G at the first position of the RCGTG motif or in the nucleotides flanking this core). Additionally, the representation of the ratio of HIFalpha ChIP-seq signal at 0.5% compared to 3% against the total (0.5%+3%) could reveal a relationship between total binding (as readout of binding affinity) and loading.

The reviewer is correct that "progressive loading" sites are essentially low affinity sites and that "early loading sites" are high affinity sites. However, affinity is often confused with overall binding intensity, so we chose to describe the sites according to the observed characteristic.

As suggested, we have examined for differences in nucleotide composition for all flanking positions within 10-bp of the core RCGTG motif and at the first position of the RCGTG motif itself. No significant difference in base composition between the progressive loading and early loading sites was observed at any position for either HIF-1 or HIF-2 sites (see below). We have added a description of this result to the revision

The suggested plots are essentially MA plots (log ratio plotted against mean signal). We had plotted these graphs, but chose not to represent them as we felt that they added little to the plots already shown in Figures 2B, 2C and 2D. We include them here for the reviewer. No significant relationship was observed between total binding and differential loading. We have added a comment to this effect in revision

- A key and important finding of this work is that HIF1 and HIF2 loading at HIF binding sites is an intrinsic property of each isoform with each isoform showing a clear and strong bias

regarding their location relative to the TSS. However, the determinants driving this selectivity are unclear. The authors show differential associations of HIF1 and HIF2 binding sites with histone modifications but these are very general marks and, as stated in the discussion, it is difficult to determine causality. Thus, it is likely that other factors determine HIF binding specificity. In this regard, is there any sequence motif significantly over-represented (depleted) in the HIF2 "only" binding sites compared with HIF1a "only"?

We have observed enrichment (over-representation) of particular transcription factor binding motifs specifically at HIF-1 sites or at HIF-2 sites. Notably, in HKC8 and RCC4 cells, HEY1/2 and ZNF263 motifs were amongst those most enriched at HIF-1 binding sites, whilst AP-1 motifs were most markedly enriched at HIF-2 binding sites. In HepG2 cells, the most enriched motifs at HIF-1 sites included SP1/2 as well as HEY2, whilst those most enriched at HIF-2 sites included FOXD2, FOXL1 and FOXC2. We have added this analysis to supplemental information.

However, in this analysis, we often see enrichment of multiple closely related motifs and it is not possible to know which, if any, of the associated transcription factors are bound. Therefore, we have also examined the overlap between HIF-1 and HIF-2 binding sites and binding of other transcription factors using publically available ChIP-seq analyses from the ENCODE Consortium. In this respect HepG2 cells are one of the best-studied cell lines with approximately 50 transcription factor ChIP-seq datasets available. We have included this analysis in Figure 6 of the revised manuscript and now include a discussion of those transcription factors that are bound preferentially at either HIF-1 or HIF-2 sites.

4. The data presented demonstrate that many sites are bound by both isoforms yet there is little or no cross-competition, suggesting that common sites could be co-occupied rather than simultaneously bound by one or the other isoform. Is there evidence supporting or ruling out this possibility? Do shared sites have a larger number of RCGTG motifs on average? How is the distribution of distances between HIF1a and HIF2a peak summits at shared sites? How are HIF beta peaks at these sites compared to those at sites bound by a single alpha isoform?

These are important questions and we have examined each in turn. In summary:

- a) **We do not see any significant difference in the number of HRE motifs at shared sites compared to sites that bind only one HIF isoform.**
- b) **Within the shared sites, the distribution of distances between the HIF1 α and HIF2 α peak summits is very tightly distributed about a median distance of less than one base pair (interquartile range: -21 bp to +23 bp). This distribution is almost identical to that seen when the summits for one replicate are compared with the other replicate for the same isoform, indicating that the observed distribution is within the precision of the assay.**
- c) **The HIF-1 β signal at shared sites shows the same profile as that at sites that bound a single isoform.**

Taken together, these findings strongly suggest that at each site, the two isoforms are binding to the same RCGTG motif. However, crystal structures of the two isoforms bound to DNA indicate that simultaneous binding of HIF-1 α /HIF1 β and HIF-2 α /HIF-1 β heterodimers is sterically prohibited. Since ChIP-seq signal is an average across a large number of cells, this suggests that at any instance, in any one cell, shared sites are occupied by a single isoform. The lack of competition between the two isoforms would be consistent with low overall occupancy (i.e. the site is unoccupied most of the time).

We have included, and added a brief comment on these findings in the revision.

5. CRISPR-mediated ablation of HIF2a results in a slight reduction of HIF1a protein. Assuming that these are clonal cell lines (it is not explicitly indicated in the manuscript), is this an anecdotal effect due to cell-to-cell variability or is it a reproducible effect seen in other cell clones? Does the ablation of one isoform affect the mRNA levels of the other?

The reviewer is correct that these are clonal cell lines and that this may underlie some of the difference. However, we only obtained one validated KO clone for each HIF isoform, so are unable to determine whether this observation is consistent across multiple clones. Based on the totality of data in the HIF field, we consider it more likely that this simply reflects cell-to-cell variability. We have commented on this in revision.

6. Regarding conservation of HIF binding sites across cell lines, it is clear that categorical classification based on peak calling underestimates overlap. The inclusion of a base-line, such as that shown in figures 1E and 1F, in the quantitative analysis shown in expanded figures 4G and 4H, could be helpful to get a better approximation to the number of overlapping sites across cell lines. From a functional perspective, it would be nice to compare the overlap between the genes nearest to HIF-bound sites across cell lines.

The base-line signal has now been included in Expanded View Figures 4G and 4H.

Cross-comments from referee 2:

Both of these suggestions in theory would expand the relevance of this study. However, in reality, they potentially pose significant theoretical and practical barriers. Gene expression analyses might indicate what HIF binding sites are transcriptionally active. However, given that HIF binding appears to affect epigenetic signatures that may also be influenced in a gene-specific manner by non-HIF factors, one could imagine an experimental result that provides lots of data, but one without any meaningful pattern, at least at this time. Nevertheless, it may be reasonable to suggest or request that the authors provide either broad transcriptional analyses or targeted rtPCR analyses in the three cell lines examined (HKC-8, RCC4, HepG2) using RNA/cDNA samples that were prepared in parallel. For targeted rtPCR analyses, it would be helpful to examine a dozen or so select genes that are similar as well as ones that differ between the three cell lines.

See comments above. In addition, we have performed RNA-seq analysis of transcript levels in normoxia and following 16 hours at 0.5% hypoxia in HKC-8 cells, HepG2 cells and RCC4 cells stably transfected with wtVHL to confer normal regulation of HIF. Genes were ranked according to hypoxic induction and Gene Set Enrichment Analysis (GSEA) was performed for genes (TSS) closest to canonical HIF-binding sites in each cell line. Consistent with our previously published findings in MCF-7 cells, HIF bound genes were strongly enriched amongst genes that were up- but not downregulated by hypoxia, confirming the functional relevance of the observed HIF-binding patterns. We have added this analysis as a new Expanded View Figure.

However, my main concern with asking for additional transcriptional, and especially more cellular, studies is the practical barrier that this would impose on the authors. ChIPseq experiments are not trivial ones in terms of time, money, and effort. Asking for additional cell line studies, in particular, would likely be a death knell for this study and would reinforce an unfortunate pattern of electronic "piling-on" that permeates the review process these days. This study has enough merit with its current content to warrant publication now, in my opinion.

Cross-comments from referee 3:

Yes, I agree. Global transcriptome analyses of HIF target gene expression could certainly help to determine the functional impact of the different DNA binding patterns of HIFs in different biological contexts. On the other hand, the confirmation of results in a second line will strengthen the conclusions. However, asking for the generation of additional CRISPR-edited cell lines implies an awful amount of work just to get a confirmatory result rather than providing novel insights. Thus, in my opinion, other (independent) works should confirm the lack of competition between both HIF isoforms. As regards of transcriptome analysis, authors could make use of published RNA-seq datasets to correlate expression with binding patterns".

See comments above. We now provide a correlative analysis of our canonical HIF-binding sites with RNA-seq analyses in HKC-8, HepG2 and RCC4 cells expressing wtVHL.

Thus, as indicated in my evaluation report and in agreement with referee #2, I believe the work has sufficient merit to be published after minor modifications addressing the points I raised before.

2nd Editorial Decision

24 September 2018

Thank you for the submission of your revised manuscript. We have now received the enclosed comments from the referees and I am happy to tell you that all support its publication now. Only a few minor changes are needed before we can proceed with the official acceptance.

Please explain what is shown and the statistics in the Box-and-whisker plots in figure 3 D-G in the figure legend.

Fig 5A + B are called out before Fig 4, and Fig 4E + G are called out after Fig 5. Please correct.

Your manuscript has 8 EV figures, but we can only offer a maximum of 6 EV figures. You could either move 1 or 2 EV figures to the main manuscript file, or combine some EV figures.

The 3 EV tables are fine, but are submitted as pdf files, and we need excel or word files. Please also include the table titles and legends in the excel or word file.

EMBO press papers are accompanied online by A) a short (1-2 sentences) summary of the findings and their significance, B) 2-3 bullet points highlighting key results and C) a synopsis image that is 550x200-400 pixels large (the height is variable). You can either show a model or key data in the synopsis image. Please note that text needs to be readable at the final size. Please send us this information along with the revised manuscript.

I look forward to seeing a final version of your manuscript as soon as possible. Please let me know if you have any questions or comments.

REFEREE REPORTS

Referee #1:

The authors have done a thorough job of responding to previous concerns.

Referee #2:

The revised report addresses many of the more reasonable concerns raised by the Reviewers 1 and 3. As such, it represents an even more complete body of work than the original submission, which in my opinion was acceptable in that state. I support publication of the current study.

Referee #3:

The authors have addressed my concerns. I have no further questions.

2nd Revision - authors' response

9 October 2018

The authors performed all minor editorial changes.

YOU MUST COMPLETE ALL CELLS WITH A PINK BACKGROUND ↓
PLEASE NOTE THAT THIS CHECKLIST WILL BE PUBLISHED ALONGSIDE YOUR PAPER

Corresponding Author Name: David Robert Mole

Manuscript Number: EMBOR-2018-46401